# Completing Linnaeus's inventory of the Swedish insect fauna: Only 5,000 species left?

**Fredrik Ronquist**[1]*, **Mattias Forshage**[2], **Sibylle Häggqvist**[2], **Dave Karlsson**[3], **Rasmus Hovmöller**[2], **Johannes Bergsten**[2], **Kevin Holston**[1], **Tom Britton**[4], **Johan Abenius**[3], **Bengt Andersson**[3], **Peter Neerup Buhl**[3], **Carl-Cedric Coulianos**[3], **Arne Fjellberg**[3], **Carl-Axel Gertsson**[3], **Sven Hellqvist**[3], **Mathias Jaschhof**[3], **Jostein Kjærandsen**[5], **Seraina Klopfstein**[6], **Sverre Kobro**[3], **Andrew Liston**[7], **Rudolf Meier**[8], **Marc Pollet**[9,10,11], **Matthias Riedel**[12], **Jindřich Roháček**[13], **Meike Schuppenhauer**[14], **Julia Stigenberg**[2], **Ingemar Struwe**[3], **Andreas Taeger**[7], **Sven-Olof Ulefors**[3], **Oleksandr Varga**[15], **Phil Withers**[3], **Ulf Gärdenfors**[16]

1 Dept. Bioinformatics and Genetics, Swedish Museum of Natural History, Stockholm, Sweden, 2 Dept. Zoology, Swedish Museum of Natural History, Stockholm, Sweden, 3 Station Linné, Ölands Skogsby, Färjestaden, Sweden, 4 Dept. Mathematics, Stockholm University, Stockholm, Sweden, 5 Tromsø University Museum, UiT—The Arctic University of Norway, Langnes, Tromsø, Norway, 6 Natural History Museum Basel, Basel, Switzerland, 7 Senckenberg German Entomological Institute, Müncheberg, Germany, 8 Lee Kong Chian Natural History Museum and Dept. Biological Sciences, National University of Singapore, Singapore, Singapore, 9 Research Institute for Nature and Forest, Bruxelles, Belgium, 10 Research Group Terrestrial Ecology, Ghent University, Ghent, Belgium, 11 Entomology Unit, Royal Belgian Institute for Natural Sciences, Bruxelles, Belgium, 12 Bavarian State Collection of Zoology, München, Germany, 13 Dept. Entomology, Silesian Museum, Opava, Czech Republic, 14 Dept. Soil Zoology, Senckenberg Museum of Natural History Görlitz, Görlitz, Germany, 15 Schmalhausen Institute of Zoology, National Academy of Sciences of Ukraine, Kyiv, Ukraine, 16 Swedish Species Information Centre, Swedish University of Agricultural Sciences, Uppsala, Sweden

* fredrik.ronquist@nrm.se

**Data Availability Statement:** All relevant data are made available in a public GitHub repo: https://github.com/ronquistlab/swedish-insect-fauna.

## Abstract

Despite more than 250 years of taxonomic research, we still have only a vague idea about the true size and composition of the faunas and floras of the planet. Many biodiversity inventories provide limited insight because they focus on a small taxonomic subsample or a tiny geographic area. Here, we report on the size and composition of the Swedish insect fauna, thought to represent roughly half of the diversity of multicellular life in one of the largest European countries. Our results are based on more than a decade of data from the Swedish Taxonomy Initiative and its massive inventory of the country's insect fauna, the Swedish Malaise Trap Project The fauna is considered one of the best known in the world, but the initiative has nevertheless revealed a surprising amount of hidden diversity: more than 3,000 new species (301 new to science) have been documented so far. Here, we use three independent methods to analyze the true size and composition of the fauna at the family or sub-family level: (1) assessments by experts who have been working on the most poorly known groups in the fauna; (2) estimates based on the proportion of new species discovered in the Malaise trap inventory; and (3) extrapolations based on species abundance and incidence data from the inventory. For the last method, we develop a new estimator, the combined non-parametric estimator, which we show is less sensitive to poor coverage of the species pool than other popular estimators. The three methods converge on similar estimates of the size and composition of the fauna, suggesting that it comprises around 33,000 species. Of

**Funding:** FR was supported by grant 2014-05901 from the Swedish Research Council (https://vr.se), DK by grant 217-2004-2101 from FORMAS, a Swedish research council for sustainable development (https://formas.se) and JR by grant MK000100595 from the Ministry of Culture of the Czech Republic (https://mkcr.cz). The Swedish Taxonomy Initiative (https://www.artdatabanken.se/en/the-swedish-taxonomy-initiative) funded the Malaise trap inventory and many of the taxonomic research projects contributing to the charting of the Swedish insect fauna since 2003. The funders had no role in study design, data collection and analysis, decision to publish, or preparation of the manuscript.

**Competing interests:** The authors have declared that no competing interests exist.

those, 8,600 (26%) were unknown at the start of the inventory and 5,000 (15%) still await discovery. We analyze the taxonomic and ecological composition of the estimated fauna, and show that most of the new species belong to Hymenoptera and Diptera groups that are decomposers or parasitoids. Thus, current knowledge of the Swedish insect fauna is strongly biased taxonomically and ecologically, and we show that similar but even stronger biases have distorted our understanding of the fauna in the past. We analyze latitudinal gradients in the size and composition of known European insect faunas and show that several of the patterns contradict the Swedish data, presumably due to similar knowledge biases. Addressing these biases is critical in understanding insect biomes and the ecosystem services they provide. Our results emphasize the need to broaden the taxonomic scope of current insect monitoring efforts, a task that is all the more urgent as recent studies indicate a possible worldwide decline in insect faunas.

## Introduction

More than 250 years after Linnaeus's pioneering attempts at charting the diversity of the planet, we still have only a vague idea about the true size and composition of faunas and floras. Current knowledge about global biodiversity is based on extrapolation from small samples to total global species richness based on questionable assumptions about ecology [1–3] or taxonomic classification [4].

In recent decades, an increasing number of inventories have tried to improve the precision of diversity estimates. However, many inventories provide limited insight because they focus on a small taxonomic or ecological subsample of a tiny geographic area [5, 6]. Tropical regions and biodiversity hotspots are popular inventory targets because of their spectacular diversity, but their faunas and floras also belong to the most challenging to characterize, and we are not even close to a complete inventory of any of these areas.

The focus on tropical diversity is based on the fact that the temperate regions have less to offer in terms of uncharted species richness. But how much do we actually know about the macroscopic floras and faunas (the multicellular organisms) of the most intensely studied corners of the planet? This is the question we address in the present paper.

Specifically, we focus on Sweden, one of the largest countries in Europe with respect to area, and its insect fauna, believed to comprise roughly half the diversity of multicellular organisms in the country. Thanks to the Linnaean legacy, the Swedish flora and fauna are among the best known in the world. Since 2002, the knowledge of Swedish organismal diversity has also increased substantially thanks to the Swedish Taxonomy Initiative [7, 8], the aim of which is to completely chart the flora and fauna of the country.

Insects have received a considerable amount of attention from the initiative. In addition to supporting taxonomic research projects on the most poorly known insect groups, the initiative has also funded a massive countrywide inventory, the Swedish Malaise Trap Project [9, 10]. Malaise traps [11] are particularly effective in collecting the insect orders Diptera (mosquitoes, gnats, midges and flies) and Hymenoptera (sawflies, wasps, ants and bees), to which most of the poorly-known insect groups belong.

At the start of the initiative, the known Swedish insect fauna was estimated to comprise 24,300 species [12]. Since then, using a combination of traditional and molecular methods, taxonomists have added 3,097 species, 301 of which have been described as new to science (S1 Table). It is clear that many more species remain to be discovered and described, challenging biologists to reexamine questions that were long thought to have been satisfactorily answered:

how large is the Swedish insect fauna really, and what is its true taxonomic and ecological composition?

Here, we address these questions using three different methods. First, we asked experts involved in the Swedish Taxonomy Initiative and the Malaise trap inventory to provide informed guesses about the number of Swedish species for their group(s). For critical groups, we then checked the expert assessments against two independent species richness estimates based on data from the Malaise trap inventory, the first based on the proportion of new species encountered, the second on extrapolations from abundance and incidence data. For the latter, we develop a new extrapolation method, the combined non-parametric estimator, which we show is less sensitive to poor coverage of the species pool than other popular estimators. Because the results from the three approaches are consistent, and the known species stock constitutes such a large fraction of the predicted total ($>$ 80%), we argue that our results provide the first clear insights into the real size and composition of a sizeable insect fauna.

We analyze the taxonomic and ecological composition of the estimated fauna, and show that it is richer in Hymenoptera and Diptera than previously thought; it also contains more decomposers and parasitoids. Using historical sources, we show that similar biases have gravely distorted our understanding of the Swedish insect fauna in the past. Finally, we compare latitudinal gradients in the size and composition of the fauna, as indicated by the Malaise trap inventory, with those of known European insect faunas. We argue that some of the discrepancies we see are caused by knowledge biases affecting our current understanding of the latter.

## Methods

### Current knowledge of the Swedish insect fauna

To assess the current knowledge of the Swedish insect fauna, we used the content of DynTaxa (http://dyntaxa.se), the official checklist of the Swedish flora and fauna. Data were pulled from DynTaxa on February 8, 2017 (data provided in a public GitHub repo, see "Data availability"). We included all species recorded as reproducing ("Reproducerande"), accidental and reproducing ("Tillfälligt reproducerande"), uncertain ("Osäker förekomst"), accidental and not reproducing ("Tillfällig förekomst, ej reproducerande"), nationally extinct ("Nationellt utdöd"), no longer reproducing ("Ej längre reproducerande") or possibly nationally extinct ("Möjligen nationellt outdöd"). Due to lack of knowledge, very few insect species occurrence records in DynTaxa are currently classified into any of these detailed categories; the species are simply assumed to be "reproducing". In well-known insect groups, a handful of species are known to be accidentals or belong to one of the other categories, but we chose to include them in the species counts to avoid systematic bias towards higher counts in poorly known groups.

Five species lacked occurrence status information in Dyntaxa: *Dixella martini* (Dixidae), *Aphis violae* (Aphididae), *Diaspis boisduvalii* (Diaspididae), *Cosmopterix pulchrimella* (Cosmopterigidae) and *Haplothrips vuilleti* (Phlaeothripidae). They were nevertheless included in our analysis as Swedish based on other data. The family placement of *Sciarosoma nigriclava* (Diptera; listed as the junior synonym *S. borealis*) is given as uncertain in Dyntaxa; it was included by us in the Diadocidiidae, following the classification in Fauna Europaea [13]. Five species in DynTaxa lack information about family classification, namely *Dimorphopterus spinolae* and *Ischnodemus sabuleti* in Blissidae, and three species of *Cymus* in Cymidae: *C. claviculus*, *C. glandicolor*, and *C. melanocephalus*. The family information was added for these records. For a few taxa, the number of Swedish species given in the DynTaxa 2017 checklist is obviously wrong and was corrected (details in S1 Table).

## Historical knowledge of the Swedish insect fauna

Information was extracted from the three most comprehensive surveys published before the start of the Swedish Taxonomy Initiative. The first [14] lists all Swedish insect species known to Linnaeus in 1761. These species were mapped to currently valid species so that they could be placed in the classification used by us. The second [15] and third [12] do not provide species lists. Instead, they summarize the knowledge of the fauna at the time (1920–1922 and 2003, respectively) at higher taxonomic levels, mostly at the family level, and give the estimated number of known species of each taxon. We mapped these numbers onto the higher taxa in our classification as detailed in the "Taxonomic composition" section below and in S1 Table. In addition, the 2003 numbers were corrected for some groups where they are obviously wrong (see S1 Table for details).

For our analyses of data from the Swedish Malaise Trap Project, we needed explicit species lists from 2003 of the target taxa. These were reconstructed with the help of experts and the literature on each group. The resulting check lists from 2003 are provided in the public GitHub repo accompanying the paper (see "Data availability").

## Known European insect faunas

Data on the European insect fauna were obtained from the Fauna Europaea database (version 2.6, distribution 3) kindly provided by Yde de Jong. This version of the database was current as of February 2017. The database includes both taxonomic data and distribution data, detailing the occurrence of each species in each of the European countries and geographic regions recognized in Fauna Europaea. The database dump we used is provided in the GitHub repo.

The Fauna Europaea data were used to analyze the latitudinal gradient in the species richness of known European insect faunas. In this analysis, we excluded all areas outside of Europe, as well as European areas of countries that mostly lie outside of Europe (Russia and Turkey). We also excluded some exotic areas with faunas that are not typical of the country they belong to: Madeira, the Selvagens Islands, the Azores, Canary Islands, and Gibraltar. Country latitude data were taken from a file provided by Google [16]. Land area data were taken from the EuroStat NUTS survey in 2013 [17] and complemented with data from Wikipedia [18]. The raw data are provided in the GitHub repo.

## Taxonomic composition

Taxonomic composition of the Swedish fauna was analyzed in terms of the families and orders recognized in the DynTaxa classification as of February 8, 2017. However, at the family level, we divided the three most species-rich insect families (Ichneumonidae, Braconidae and Staphylinidae) into subfamilies to make the family-level units more comparable in size. When data on the Swedish fauna were analyzed from older sources using a less detailed classification [14, 15], we split the listed species numbers based on the available literature from the time (see also S1 Table). The analysis of the European data, and of taxa expected to occur in Sweden, was based on the Fauna Europaea taxonomy [13].

## Ecological composition

In the analysis of life-history traits, we focused on two traits that are conservative enough that they can be reasonably assumed, in most cases, to be homogeneous within the family-level groups we used: the main feeding niche and the main feeding (micro-)habitat. This is the niche and habitat of the immature stages (the main feeding stages), and may or may not be the same as the niche and habitat of more short-lived adult stages. Data were taken from standard

works [15, 19–23] complemented with data from relevant taxonomic specialists and the primary literature.

Specifically, we defined the feeding niches as follows:

*Parasite*. This includes bloodsuckers, endoparasites (botflies) and exoparasites (lice).

*Phytophage* (plant feeder). This includes both chewers and sap suckers, as well as stem borers, leaf miners, root feeders and gall inducers.

*Phytophage-parasitoid*. This is restricted to all primary parasitoids of plant feeders.

*Predator*. This is restricted to free-living predators, it does not include parasitoids.

*Predator-parasitoid*. This includes both primary parasitoids of predators and all hyperparasitoids (parasitoids of parasitoids).

*Saprophage* (decomposer). This includes scavengers, decomposers, fungivores and microflora/bacterial grazers.

*Saprophage-parasitoid*. This is restricted to all primary parasitoids of decomposers.

The (micro-)habitats of the main feeding stages (larvae, nymphs) were defined as follows:

*Fungi*. Within fruiting bodies of macrofungi.

*Homeothermic animals*. On homeothermic animals (birds, mammals). The habitat of parasites (bloodsuckers, endoparasites and exoparasites).

*Plants*. In or on aerial parts of living plants, excluding wood.

*Soil*. In soil, including ground litter as well as living roots.

*Temporary habitats*. In or on ephemeral resources of rich nutrients, such as bird nests, carrion, dung and fermenting sap. Also includes indoors habitats.

*Water*. In or on water.

*Wood*. In wood.

Groups were coded for the dominant trait; notable exceptions and other comments are given in the tables. Because of different taxon or group circumscriptions, the traits differ in a few cases between the Swedish fauna (S1 Table), the groups used in analyzing inventory data (S4 Table), and the European fauna (S6 Table).

## Expert estimates of the true fauna

Expert estimates of the total size of the Swedish fauna were based on the opinions of taxonomic specialists with intimate knowledge of the Swedish insect fauna. They were obtained for all family-level groups in the fauna at two occasions: in 2007, before the Malaise trap inventory material was processed, and in the spring of 2017 (current estimate). The 2007 estimates were based on less extensive surveys of experts, and we only give them here for the target groups used in the species richness analysis of the inventory data (S5 Table). The estimates from 2017 are detailed in S1 Table, together with the experts responsible for them.

For a few groups, we were not able to find local specialists willing to estimate the total size of the fauna. In those cases, MF, RH and FR estimated the total size of the Swedish fauna as follows. For small well-known groups (Diplura, Archaeognatha, Zygentoma, Dermaptera), the current number of known species was accepted as the estimate of the true fauna except in the case of Dermaptera, where we added one expanding species known from nearby countries

(*Labidura riparia*). Protura are relatively well-known but rarely studied from a faunistic or systematic viewpoint, and are thus likely to contain undiscovered species. We estimated the true fauna to be 30% larger than the currently known one (an increase by a factor of 1.3).

For the two main groups of vertebrate parasites, Phthiraptera and Siphonaptera, we based our estimates on each family's host preferences. Families of parasites exclusively or mainly on man and domestic mammals were considered well-known and we accepted the current number as final. For families of parasites on wild rodents, insectivores, bats and carnivores, plus siphonapteran parasites on birds, we increased current numbers by a factor of 1.4. For phthirapteran families including parasites of birds we used a factor of 1.6. This seemed reasonable considering the patterns of host range, the fauna of hosts in Sweden, the fauna of these families in better studied neighboring countries, and the relatively poor global taxonomic knowledge of phthirapteran bird parasites. In two species-poor families where these considerations suggested no addition, we nevertheless added expected species that are known from neighboring countries (one each in Enderleinellidae and Hoplopleuridae).

## Malaise trap inventory

For critical groups, we checked the expert assessments against two independent species richness estimates based on data from the Swedish Malaise Trap Project. The project is the first systematic inventory of the whole diversity of the Swedish insect fauna, and it particularly targets the most poorly known groups in the orders Diptera and Hymenoptera [9, 10]. The material was collected using 73 Malaise traps, deployed at different sites throughout the country and representing a wide range of habitats. Most of the traps were run continuously from 2003–2006, but some traps were run for only part of this period, and a couple of traps were run in the period 2007–2009. Detailed data on the trap sites and the samples collected (1,919 samples in total) are provided in S2 Table (see also http://www.stationlinne.se/en/research/the-swedish-malaise-trap-project-smtp/). The trap sites were classified into four biogeographic regions and six habitat classes to simplify analyses of the data (S2 Table).

The inventory material (unsorted material and identified specimens) is deposited at the Swedish Museum of Natural History (http://www.nrm.se) through its collaboration with the Station Linné Field Station (http://stationlinne.se). The material is continuously sorted into more than 300 taxonomic fractions, mostly at the family or subfamily level (see http://www.stationlinne.se/en/research/the-swedish-malaise-trap-project-smtp/taxonomic-units-in-the-smtp/), which are sent out to experts for identification. The returned data are validated and cleaned, and identifications are matched to the national checklist (DynTaxa) as far as possible. Data on the sorted fractions is available in a separate dataset at the Global Biodiversity Information Facility portal (https://www.gbif.org), at the Swedish natural history collection portal (https://naturarv.se) and at the Swedish biodiversity data hub (https://bioatlas.se). For access to the material and for questions about the data, contact DK or JB.

Here, we analyze data from the first 165,000 specimens that have been identified, representing about 1% of the total catch [10]. The data analyzed here represent all the cleaned data files that were available in the spring of 2017 (S3 Table). In total, there are 127 files, most of which contain abundance data (117 files), but some of which only have incidence data (10 files). The incidence data include several groups (Braconidae: Cheloninae, Braconidae: Rogadinae, Cecidomyiidae: Porricondylinae (*sensu lato*) and Psylloidea) for which abundance data were not reported consistently for all samples and species. The files are provided in the Data Package in the Supporting Information.

We divided the data into 79 datasets, each corresponding to a set of samples of a particular taxon or assemblage of taxa ("analysis taxon") processed by a different expert or group of

experts (S4 Table). The data are available in the data files provided as Supporting Information. The analysis taxa mostly correspond to families or subfamilies of Diptera and Hymenoptera, and to orders or major portions of orders for other insect groups.

The analyzed information is estimated to represent from 2 to 67% of the entire catch of each analysis taxon (S4 Table). In total, the abundance data comprise 157,225 specimens (30,643 records) identified to 3,916 taxa. The incidence data represent another 1,516 observations of 279 taxa. The incidence data roughly corresponds to 7,800 specimens, if this material contained the same average number of specimens per species and sample as the material covered by the abundance data.

Specimens or records that were determined only to genus level (marked by "sp." in the Species field in the data file provided in the Supporting Information) were removed prior to analyses of species richness and compositional trends. This affected 14,034 specimens (161 taxa) in the abundance data (most of these are females of Phoridae, which cannot at this point be identified to species) and 36 records (8 taxa) in the incidence data. In a few cases, determinations were to unspecified species that may or may not be distinct (affecting 40 records of abundance data marked by "sp. indet." in the Author field); these records were included in the analyses under the assumption that they represent distinct species.

**Species richness estimates.** To estimate the total Swedish fauna of the target groups from Malaise trap inventory data, we used two different approaches. The first approach was based on the proportion of new species encountered in the inventory, using a method analogous to mark-recapture estimation of population size. In the simplest form, the **mark-recapture estimator** (**MR**) simply uses the fraction of new ('unmarked') species encountered in a sample to extrapolate the number of known species (prior to sampling) to the total number of species in the species pool. This version is known as the Lincoln-Petersen method [24] or the Petersen-Lincoln index [25]. Unfortunately, this estimator is biased when the sample size is small, and it is undefined when no species of the group are known prior to sampling.

These problems are addressed by the **Chapman version of the MR** estimator [26], which is defined as

$$\hat{S}_{MR} = \frac{S_{obs} + 1}{S_{obs,known} + 1} \times (S_{known} + 1) - 1, \tag{1}$$

where

$S_{obs}$ = number of observed species,

$S_{obs,known}$ = number of observed species known previously, and

$S_{known}$ = number of known species.

The Chapman version of the MR estimator was implemented in an R function, which is provided in the public GitHub repo accompanying the paper (see "Data availability").

The second approach was based on extrapolations from the abundance and incidence data of the inventory. For these extrapolations, we used species richness estimators implemented in the R packages vegan, version 2.4–3 [27], BAT version 1.6.0 [28, 29] and in custom R functions (provided in the GitHub repo). There are two types of non-parametric estimators provided by vegan [see also 30]. The first set is based on abundance data, and include the estimators commonly referred to as **Chao1** (called abundance-based Chao in the vegan documentation) and **ACE** (the **abundance-based coverage estimator**). These estimators are provided by the vegan function 'estimateR'. The second set is available through the vegan function 'specpool', and include the estimators commonly referred to as **Chao2**, the **first-order jackknife** (**Jack1**), the **second-order jackknife** (**Jack2**) and the **bootstrap** (**Boot**). The second set of estimators is based on the incidence of species at the sample sites and can be used both for abundance and

incidence data, whereas the first set of estimators is only applicable to abundance data. The vegan package also provides a **parametric species richness estimator** (**Preston**) based on the Preston model through the functions 'prestonfit' and 'veiledspec'. It requires abundance data, which are fitted to a lognormal model. See the vegan documentation for a discussion of these estimators and how they are implemented in vegan, with references to the original literature.

To address some of the shortcomings of current estimators, which became obvious in the course of our study, we developed a new non-parametric estimator, which we call the **combined non-parametric estimator** (**CNE**). It is based on the idea of combining the Chao1 and Chao2 extrapolations. The Chao1 extrapolation is based on the number of singleton and doubleton specimens found in the total sample [31]. This might be expected to estimate the fauna of the sample sites more accurately than the total fauna, which is particularly problematic when trying to estimate the fauna of a large and heterogeneous area from a sample coming from a small number of sites. In contrast, the Chao2 extrapolation is based on the number of species that occur at only one sampling site (singleton species) or at two sampling sites (doubleton species) [32]. Thus, Chao2 explicitly takes the heterogeneity across sites into account, which should result in a better estimate of the total fauna under these conditions.

In the Chao2 case, however, it is possible that we have underestimated the number of singleton and doubleton species because of undersampling of the selected sites. That is, more extensive sampling at the same sites would potentially result in the discovery of more of the highly localized singleton and doubleton species occurring there. Of course, more extensive sampling might also reduce the number of singleton and doubleton species because some species may turn out to be more widespread than indicated by the original sample. If many of the real singleton and doubleton species are rare at the sites where they occur, then it might be reasonable to assume that the proportion of singleton and doubleton species observed at a site would remain the same even with more extensive sampling there.

This leads to an estimator applying a Chao1-type extrapolation to obtain an estimate of the number of singleton and doubleton species one would have observed with more extensive sampling of the selected sites. The estimated numbers of singleton and doubleton species can then be used to estimate the size of the total fauna using Chao2-type extrapolation.

Formally, define

$S^{(i)}$ = number of species at site $i$,

$S_{obs}^{(i)}$ = number of species observed at site $i$,

$S_1^{(i)}$ = number of species that only occur at site $i$,

$S_2^{(i)}$ = number of species that only occur at site $i$ and one other site,

$S_{obs}^{(i)}(1)$ = number of species observed at site $i$ and only there,

$S_{obs}^{(i)}(2)$ = number of species observed at site $i$ and one other site,

$F_1^{(i)}$ = number of observed species at site $i$ that are only observed once there,

$F_2^{(i)}$ = number of observed species at site $i$ that are only observed twice there,

$Q_1$ = number of species that only occur at one site, and

$Q_2$ = number of species that only occur at two sites.

We now have the following estimators:

$$\hat{S}^{(i)} = S^{(i)}_{obs} + \frac{F^{(i)}_1(F^{(i)}_1 - 1)}{2F^{(i)}_2 + 1} \tag{2}$$

estimates the number of species at site $i$ (this is the same as the Chao1 estimator),

$$\hat{S}^{(i)}_1 = S^{(i)}_{obs}(1) + \frac{F^{(i)}_1(F^{(i)}_1 - 1)}{2F^{(i)}_2 + 1} \times \frac{S^{(i)}_{obs}(1)}{S^{(i)}_{obs}} \tag{3}$$

estimates the number of unique species at site $i$, and

$$\hat{S}^{(i)}_2 = S^{(i)}_{obs}(2) + \frac{F^{(i)}_1(F^{(i)}_1 - 1)}{2F^{(i)}_2 + 1} \times \frac{S^{(i)}_{obs}(2)}{S^{(i)}_{obs}} \tag{4}$$

estimates the number of species occurring there and at one other site.

Since

$$Q_1 = \sum_{i=1}^n S^{(i)}_1 \tag{5}$$

and

$$Q_2 = \frac{1}{2}\sum_{i=1}^n S^{(i)}_2, \tag{6}$$

we obtain the estimators

$$\hat{Q}_1 = \sum_{i=1}^n \hat{S}^{(i)}_1 \tag{7}$$

and

$$\hat{Q}_2 = \frac{1}{2}\sum_{i=1}^n \hat{S}^{(i)}_2. \tag{8}$$

This gives us the final species richness estimator (with correction for small samples)

$$\hat{S}_{CNE} = S_{obs} + \frac{\hat{Q}_1(\hat{Q}_1 - 1)}{2\hat{Q}_2 + 1}. \tag{9}$$

Most species richness estimators based on extrapolation are known to underestimate diversity when the sample is small [33]. A recent paper [34] develops an empirical correction for this bias in non-parametric species richness estimators. These corrected estimators are provided in the BAT package. Specifically, we explored the estimators referred to as **Chao1P**, **Chao2P**, **Jack1inP** and **Jack2inP** in BAT. They correspond to the Chao1, Chao2, Jack1 and Jack2 estimators of vegan. The corrected estimators were computed using the 'alpha.accum' function of the BAT package.

## Performance of species richness estimators

To assess the performance of the species richness estimators, we used test data from two sources. First, we used well-studied groups, defined as those groups where the inventory has not discovered any new species (S4 Table). As the 'true' diversity for these groups, we used the

number of species in the current expert estimate, which was typically very close to if not identical to the number of known species (S4 Table). As a second source of test data, we used the subset of species known in 2003 in the more poorly studied groups, that is, those groups where the inventory did discover new species (S4 Table). Specifically, the abundance or incidence data for the known species in those groups were used to estimate the total number of known species in the same groups. We detected no qualitative difference between these two data sources (well-known groups versus known species in poorly studied groups), except that the first set typically included a smaller number of specimens, and the sample represented a much smaller fraction of the total species pool. We only included groups with more than 100 specimens observed from more than 5 sites in the analyses. The accuracy of the estimators was measured using the squared error and bias of the log estimates, that is, $(\log \hat{S} - \log S)^2$ and $\log \hat{S} - \log S$, respectively, where $\hat{S}$ is the estimate and $S$ the true value.

## Statistical analyses and visualization

For statistical analyses and for generating the plots shown in the paper, we used standard functions in R, version 3.3.1 [35]. The R scripts are available in the Data Package of the Supporting Information.

## Results

### Size of the fauna

The experts estimated that the Swedish insect fauna consists of approximately 33,000 species (S1 Table). This exceeds the 2003 estimate of the known fauna by 8,600 species and suggests that there are still around 5,500 species that remain to be discovered or described. Those species are concentrated, however, to a small number of taxa. Only ten of the 663 family-group taxa are expected to contain more than 100 of these missing species; only two of these taxa (Ichneumonidae: Cryptinae and Pteromalidae, both Hymenoptera) still await attention from taxonomists funded through the Swedish Taxonomy Initiative.

The identified specimens from the Malaise trap inventory (Fig 1) fall into 4,026 species, of which 689 have been putatively identified as new to science. Of the latter, only 87 have been described so far, forming part of the 301 new species to science recorded from Sweden since the start of the Swedish Taxonomy Initiative (S1 Table). A geographic and habitat breakdown of the inventory data for some key taxa of Diptera and Hymenoptera shows that many of the new discoveries, including species new to science, are made in mixed forests in the boreal and boreonemoral zones (Figs 1 and 2).

Of the inventory data we analyzed, the bulk (35 groups; 120,530 specimens) consists of abundance records for poorly known groups, where taxonomists have found new Swedish species in the material. Together, these groups account for 29% (2,525 of 8,655 species) of the fauna that experts assume is present in the country but that had not been recorded in 2003 (Table 1). Using the ratio of new to previously known species in the catch of these groups to estimate total species richness (MR estimator, analogous to mark-recapture approximation of population size) produces results that are generally in line with expert guesses for these groups (Tables 1 and S5).

Before applying species richness estimators based on abundance and incidence data from the inventory, we tested the accuracy of select methods using data for well-known groups, and for the species pool known in 2003 of poorly known groups (S5 Table). In both cases, we thus know the total number of species in the pool, which allows us to compute the accuracy of the estimates. The tests show that estimates tend to become more accurate as the number of

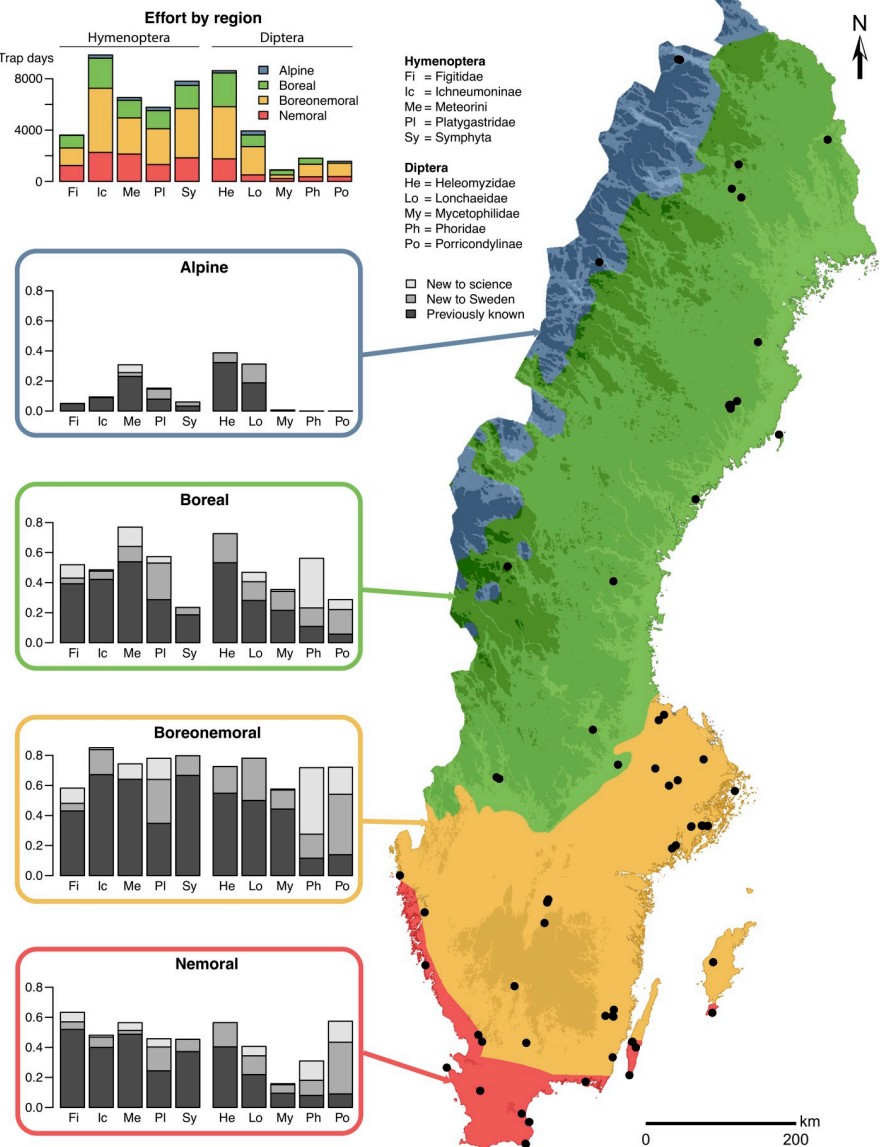

**Fig 1. Results of the Malaise trap inventory by biogeographic region for ten poorly known groups of Hymenoptera and Diptera.** The species new to science, new to Sweden, and known previously in each biogeographic region are shown as a fraction of the total species pool of the group encountered in the inventory. The dots on the map mark trap sites. Sampling effort is given as the number of trap days represented by the processed samples for each taxon and region. Many of the new species discovered in the inventory are found in the boreal and boreonemoral zones. For details on the circumscription of each group, see S4 Table.

specimens (Fig 3A), sites (Fig 3B), or specimens per site (Fig 3C) increases. However, it is the coverage (proportion of the species pool that has been sampled) that largely determines accuracy (Fig 3D): it is only at relatively high coverage (50–60% or more of the total number of species in the pool) that the estimates become reasonably robust. When half the species have been sampled, one can expect a worst-case squared error of the log estimates of approximately 0.4, which means that the estimate may be off by a factor of 1.9 (close to half or double the real species number). The average estimate is considerably better, however.

Chao1 and Chao2 consistently underestimate species diversity unless the coverage is very high, while CNE does better in terms of bias at the cost of increased variance (Figs 3–4). All

**Open meagre damp habitat**

**Forest: nemoral deciduous forest**

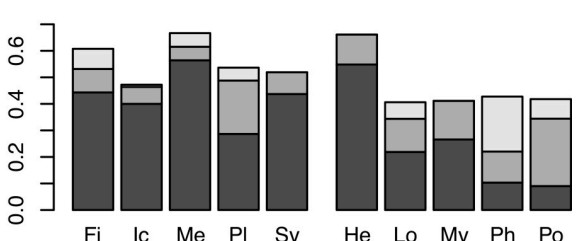

**Open meagre dry habitat**

**Forest: boreal deciduous forest**

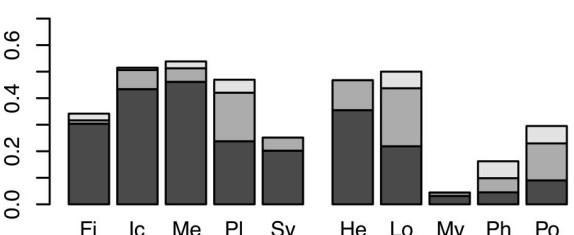
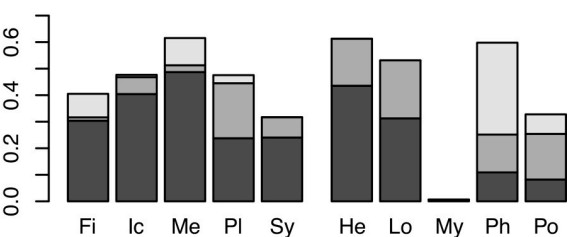

**Open lush habitat**

**Forest: coniferous forest**

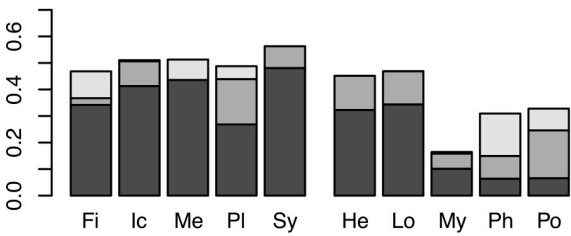
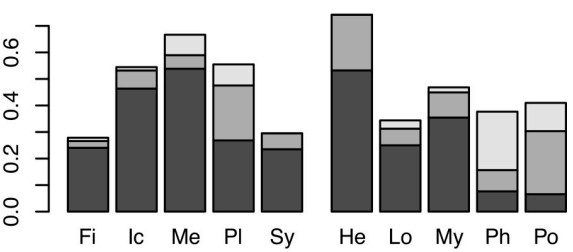

**Effort per habitat**

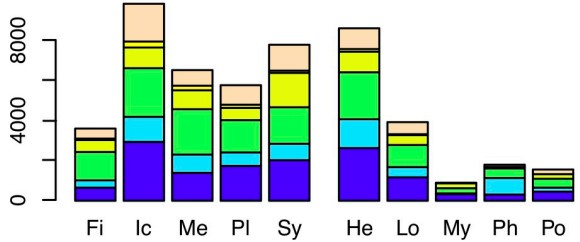

**Fig 2. Species found in the Malaise trap inventory by habitat.** Data are given for the same taxonomic groups as those in Fig 1. See S4 Table for detailed circumscription of each analysis group. Many of the new species are found in forest habitats.

**Table 1. Species richness estimates based on abundance data for poorly known insect groups from the Malaise trap inventory.** "Known spp 2003" is the number of species known from Sweden prior to the inventory. For the inventory material, we give the proportion of the catch (Prop) and the number of specimens (N) processed, as well as the total number of species found (Spp), and the number of those species that were not recorded as Swedish prior to the inventory (New). Species richness estimates include expert guesses (Expert) from 2002, before the inventory (first number) and from 2017, after partial results were known (second number). Estimates also include extrapolations using mark-recapture (MR), Chao's estimator based on the entire material (Chao1) or on occurrences of species across sites (Chao2); and the new combined non-parametric estimator introduced here (CNE). See S5 Table for more detailed results and additional species richness estimates. Data for Ichneumoninae do not include the tribe Phaeogenini.

| Taxon | Known spp 2003 | Malaise trap inventory catch | | | | Species richness estimates | | | | |
|---|---|---|---|---|---|---|---|---|---|---|
| | | Prop | N | Spp | New | Expert | MR | Chao1 | Chao2 | CNE |
| Dolichopodidae | 314 | 26% | 43011 | 204 | 31 | 356–360 | 370 | 219 | 235 | 246 |
| Phoridae | 182 | 6% | 33797 | 863 | 743 | 1100–1100 | 1306 | 1171 | 1639 | 1982 |
| Drosophilidae | 59 | 23% | 7405 | 39 | 4 | 66–80 | 66 | 46 | 63 | 106 |
| Ichneumoninae (part) | 325 | 33% | 5659 | 235 | 50 | 480–490 | 413 | 307 | 312 | 373 |
| Heleomyzidae + Odiniidae | 54 | 20% | 4911 | 62 | 21 | 62–80 | 82 | 71 | 73 | 84 |
| Sepsidae | 34 | 22% | 3870 | 16 | 1 | 32–38 | 36 | 16 | 16 | 16 |
| Platygastridae (s. str.) | 124 | 15% | 2971 | 164 | 92 | 250–270 | 282 | 186 | 218 | 267 |
| Symphyta | 584 | 28% | 2912 | 183 | 34 | 875–799 | 717 | 238 | 256 | 318 |
| Diplazontinae | 49 | 33% | 2788 | 63 | 24 | 80–90 | 79 | 68 | 69 | 73 |
| Pimplinae | 127 | 27% | 1521 | 64 | 9 | 150–150 | 148 | 112 | 97 | 121 |
| Thysanoptera | 119 | 6% | 1528 | 34 | 1 | 117–186 | 123 | 54 | 65 | 113 |
| Euphorinae: Meteorini | 37 | 27% | 1304 | 39 | 10 | 60–53 | 50 | 42 | 42 | 43 |
| Piophilidae | 17 | 26% | 1063 | 11 | 1 | 17–25 | 19 | 11 | 11 | 11 |
| Sciomyzidae | 77 | 16% | 1021 | 58 | 3 | 85–83 | 81 | 61 | 93 | 140 |
| Other groups | 888 | 2–67% | 6769 | 474 | 148 | 1358–1681 | 1289 | 627 | 887 | 1268 |
| *TOTAL SAMPLE* | *2990* | *2–67%* | *120530* | *2509* | *1172* | *5088–5515* | *5610* | *3234* | *NA* | *NA* |
| **GROUP TOTALS** | **2990** | **2–67%** | **120530** | **2509** | **1172** | **5088–5515** | **5057** | **3228** | **4076** | **5159** |

examined estimators are biased downwards, that is, they underestimate the size of the fauna, especially when a small proportion of the species pool is sampled (Fig 4). The bias is less severe for the Preston and CNE estimators, but this comes at the expense of increased variance. The estimators that take heterogeneity across sites into account (Chao2 and CNE) do considerably better than the estimators that do not (Chao1 and Preston) when a small fraction of the species pool is sampled.

The empirical correction proposed in [34] is based on the sampling effort, specifically the proportion of singletons and the number of specimens per species for estimators based on abundance data for the total sample. The corresponding measures of sampling effort for site-based data are the proportion of uniques, that is, the proportion of species that are encountered at a single site, and the number of site occurrences per species. These factors do correlate with the accuracy of estimators (S1 Fig), but not as strongly as the coverage of the species pool (Fig 3D). Even at high sampling intensities (the extreme right of the x axes in S1 Fig), the species richness estimates can be quite misleading. These are conditions when the corrected versions of the estimators are similar or identical to the original ones. There is actually a tendency for the non-parametric estimators to do worse under these settings, while there is no such trend for the parametric Preston estimator.

In terms of bias as a function of coverage of the species pool, the corrected Chao1 estimator based on abundance data for the entire sample does not do much better than the corresponding uncorrected versions (S2 Fig). However, the corrected estimators (Chao1P, Jack1P, Jack2P) based on site occurrence data show a slight but distinct improvement over the uncorrected versions (compare S2 FigB with Fig 4B).

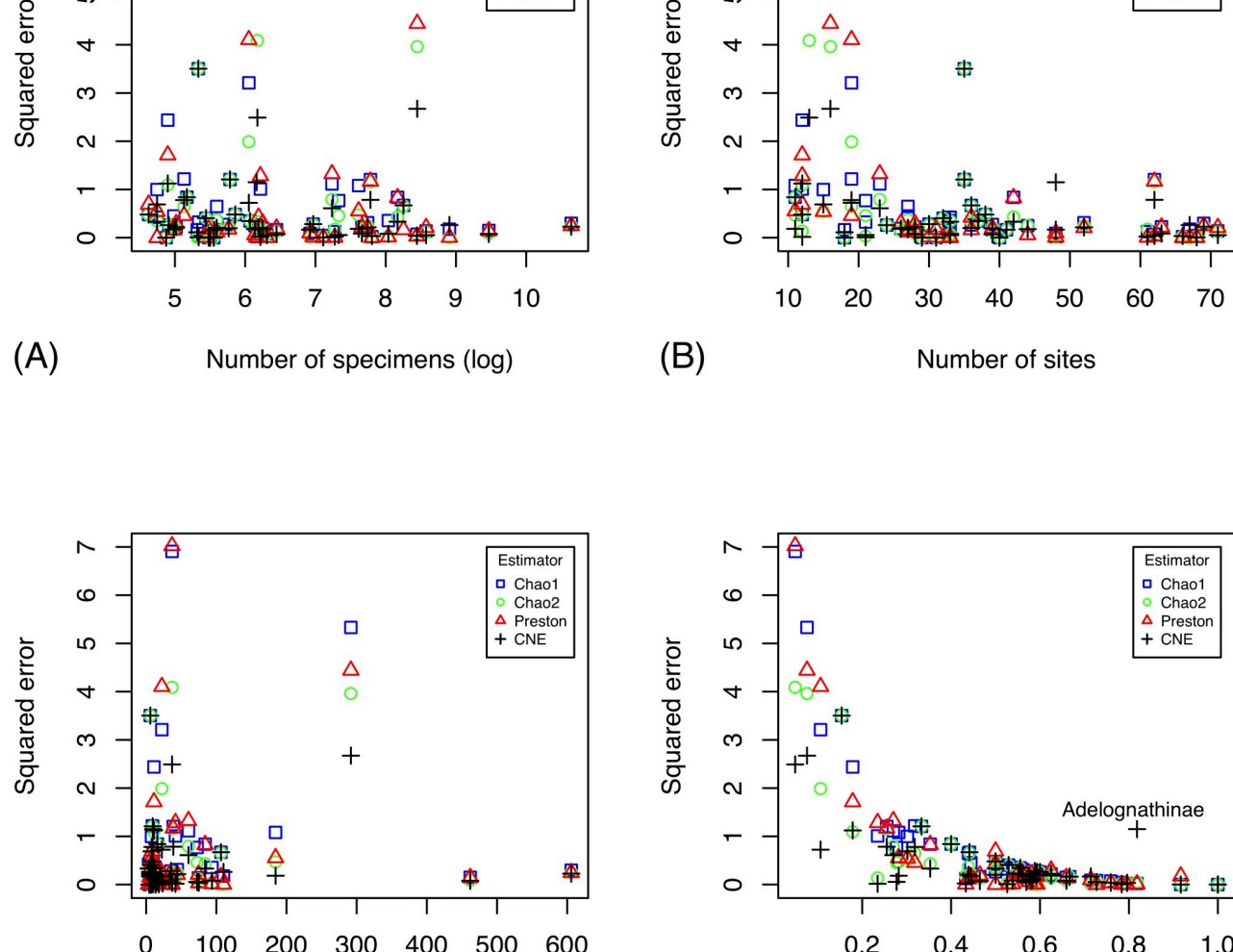

**Fig 3. Accuracy of different estimators in predicting the species richness of well-studied groups, and the species pool known in 2003 of more poorly studied groups.** The plots show the accuracy (measured as squared error of log estimates) of the species richness estimates as a function of the number of specimens (A), the number of sites (B), the number of specimens per site (C), and the fraction of the species pool sampled (D). The CNE outlier in (D) is Ichneumonidae: Adelognathinae, one of the groups with the smallest number of specimens per site.

The poorly-known groups in the inventory are represented by six times as many specimens as the well-known groups, and the inventory sample covers 50% or more of the estimated fauna (according to expert assessments) in many cases (S5 Table and Fig 5), suggesting that species richness estimates may be reliable or at least indicative of true species richness for these groups. The CNE estimates (Table 1; see also S5 Table) also tend to be in line with expert guesses and mark-recapture estimates, although there is considerable variation across groups. Conspicuous outliers are associated primarily with groups where the inventory sample does not cover the expected fauna well (S5 Table). Interestingly, the corrected versions of the estimators based on site incidence data come close to the CNE estimates, particularly Chao2P and

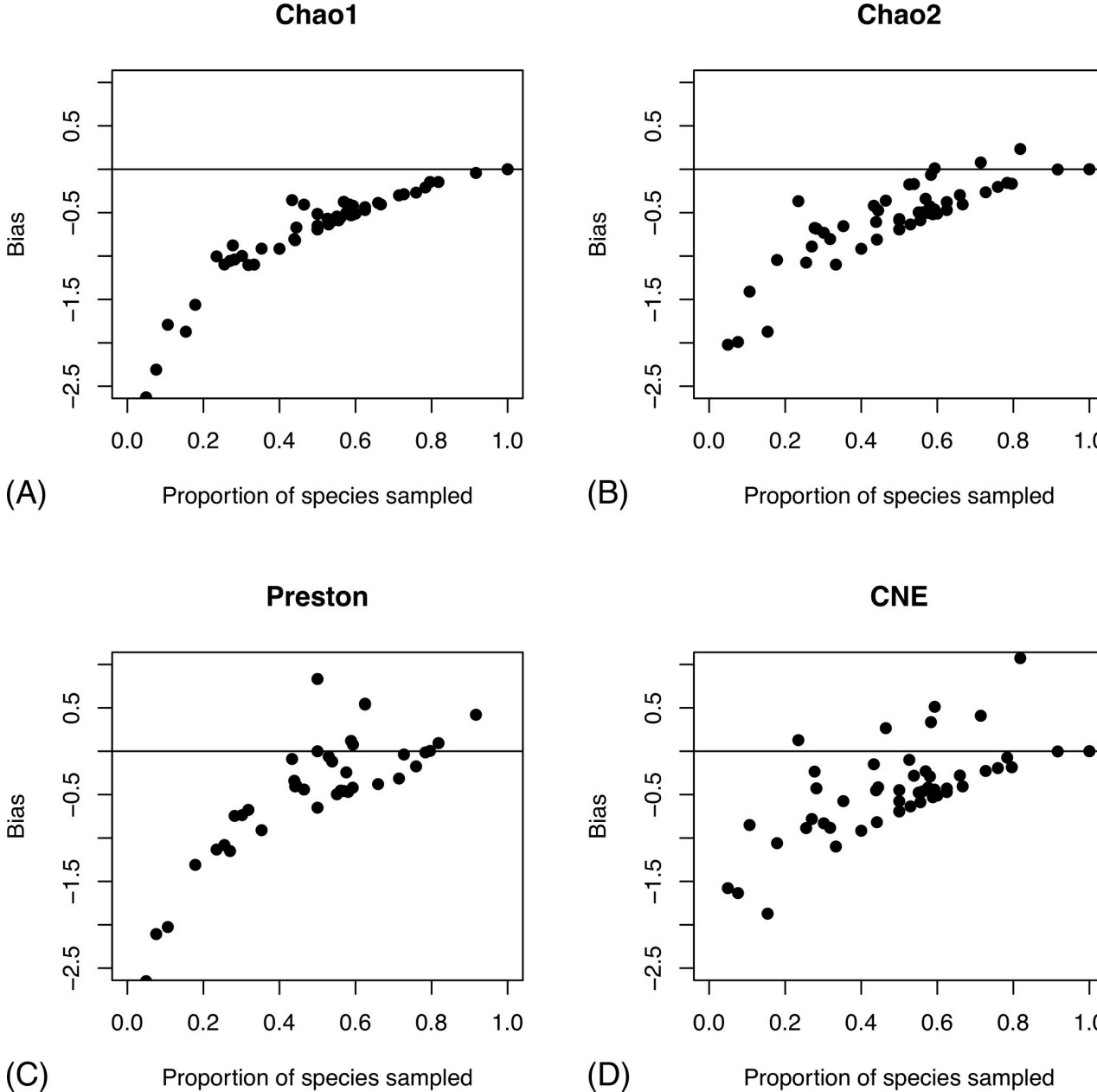

**Fig 4. Bias in species richness estimators as a function of the fraction of the species pool sampled.** We show the bias in four species richness estimators: Chao1 (A), Chao2 (B), Preston (C) and CNE (D). Bias is measured on the log scale; the horizontal line represents absence of bias.

Jack2P. The corrected Chao1p estimates, however, are not very different from the Chao1 estimates; both appear to significantly underestimate the species pool (S5 Table).

## Composition of the fauna

Given the good agreement between the expert assessments and the estimates based on the inventory data for many of the critical groups, we assume in the following that the real fauna corresponds to the expert assessments. What can we then say about its composition? Taxonomically, the Swedish fauna is dominated by Hymenoptera and Diptera (Figs 6 and 7 (left)), both orders dwarfing the Coleoptera (beetles) and Lepidoptera (butterflies and moths). There

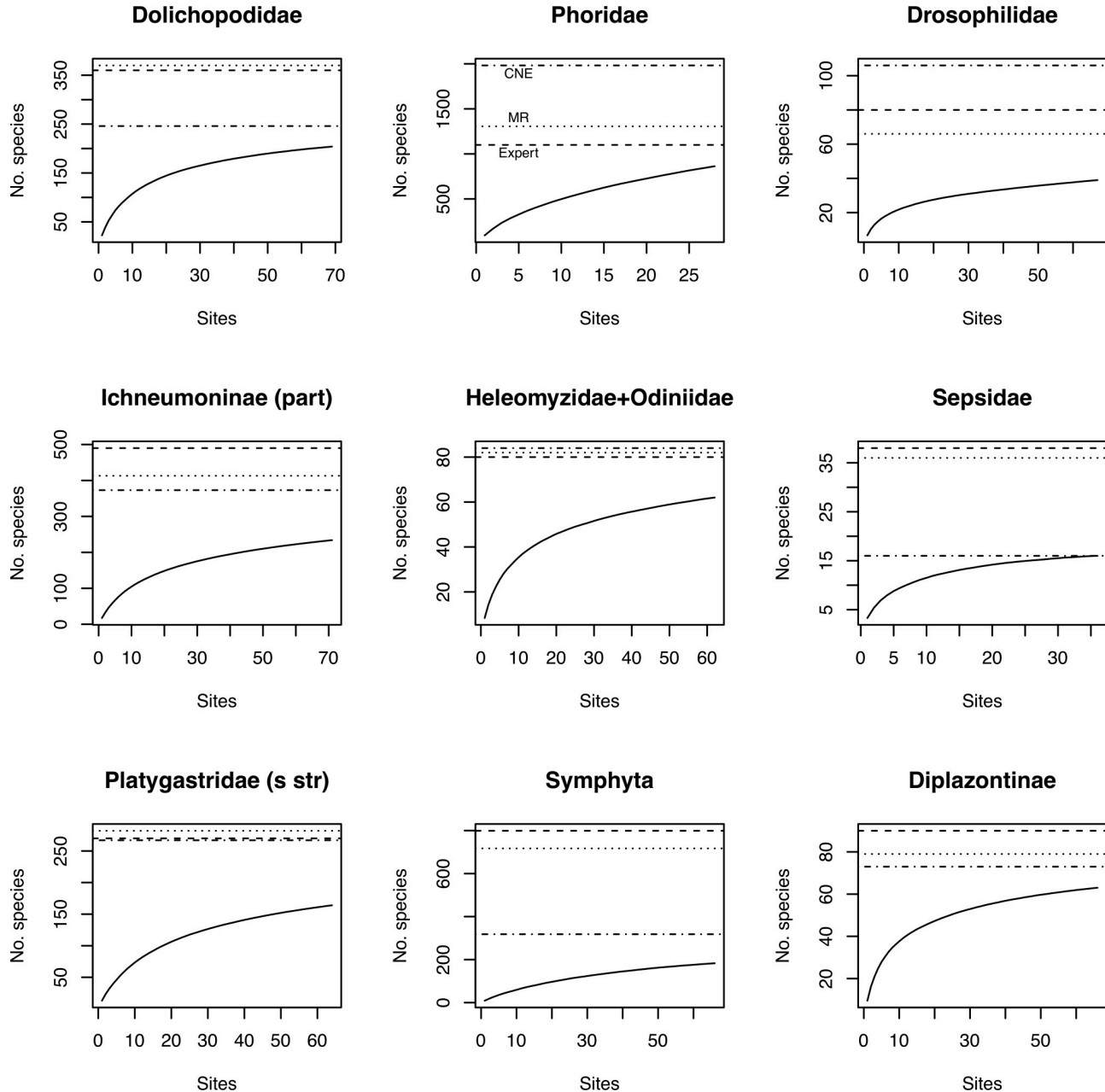

**Fig 5. Species accumulation curves for some groups studied in the Malaise trap inventory.** Accumulation curves are shown for the nine groups with the largest number of identified specimens (Tables 1 and S5). Accumulation curves are based on 10,000 random perturbations of the accumulation order of sites. The horizontal lines correspond to the expert guess from 2017 (Expert, dashed line), the estimate from the mark-recapture method (MR, dotted line) and the estimate from the combined nonparametric method (CNE, dashed and dotted line). Major disagreement among estimates tends to be associated with groups that are not covered well by the sample. In the Phoridae, the low coverage may be partly due to the small fraction (6%) of the material processed; in other cases, like the Symphyta (30% of the material processed), it appears instead that it is due to the inefficiency of Malaise traps in collecting these groups. DNA barcoding indicates that the accumulation curve for Phoridae is too steep because aberrant male-genitalia variants are sometimes misclassified by taxonomists as representing separate species [36]; this is likely to partly explain the poor performance of species richness estimators for this group.

are more than twice as many species of parasitic wasps alone as there are beetle species (Fig 6). With respect to ecological composition, our results indicate that about half of the fauna consists of phytophagous species and their parasitoids, while the decomposers and their parsitoids constitute a third (Fig 7). The remaining sixth largely consists of predators and their

parasitoids. The species of plant feeders on average host nearly one parasitoid species each, while the decomposer and predator guilds are attacked by far fewer parasitoids. In terms of main feeding microhabitat, the fauna is dominated by species associated with plants (Fig 7).

## Historical trends in the knowledge of the fauna

Analysis of the historical records show that the knowledge of the fauna has always been worse than biologists realized at the time. The knowledge has also been strongly biased both taxo-nomically and ecologically. Less than 5% of the true fauna was known to Linnaeus [18], mainly conspicuous insects living on plants, such as butterflies, moths and herbivorous beetles (Fig 8). During the 20[th] century [11, 19], as the fauna became better known, the proportion of Hyme-noptera and Diptera increased dramatically, as well as the proportion of decomposers and parasitoids.

These trends are quite noticeable even when comparing the fauna known in 2003 with the real fauna estimated here (Fig 7 (right)). The new species belong to a large extent to the Hyme-noptera and Diptera. Many of them are predicted to be decomposers or parasitoids, and more often associated with fungi or temporary habitats than other insect species. Thus, the new spe-cies substantially change our understanding of the composition of the fauna (see also Fig 6)

## Latitudinal gradients in size and composition

If we assume a power law relating area to species diversity, then the Swedish Malaise trap inventory data show a decline in diversity from the nemoral to the boreal zone. Specifically, the logarithm of the number of species encountered in the inventory divided by the logarithm of the area of the corresponding biogeographic region decreases from 0.8 over 0.7 to 0.6 through the nemoral to boreal transition. There is also a significant increase in the proportion of Diptera species and in the proportion of decomposers with increasing latitude, but there is no such trend for Hymenoptera or parasitoids (Fig 9).

The patterns in known European insect faunas are partly different. As expected, the species richness of national faunas is determined to a large extent by the land area of the country, as shown by a log–log regression analysis (Fig 10A). The residuals from this regression (Fig 10B) reveal that there are three outliers—Belarus (BY), San Marino (SM) and Iceland (IS)—that all have unexpectedly small insect faunas given their land areas. This may be because the fauna is little known (or poorly recorded in Fauna Europaea), or because of other factors. Controlling for the area effect reveals a significant but surprisingly weak latitudinal gradient in species rich-ness (Fig 10C), and this effect disappears completely when all three outliers are removed (Fig 10D), or when just the most extreme outlier, Iceland, is omitted (plot not shown; $r^2 = 0.03$, $p = 0.31$).

The Fauna Europaea data reveal a significant northern increase in the proportion of Diptera species (Fig 11A) and in the proportion of decomposers (Fig 11B), as indicated also by the Swedish Malaise trap inventory data (Fig 9). However, the European data also support a north-ern increase in the proportion of Hymenoptera species (Fig 11C and parasitoid species (Fig 11D). There is no suggestion of such trends in the Swedish data (Fig 9).

## Discussion

Estimating species richness using statistical methods is notoriously difficult. One problem is that species pools are dynamic over time. Species come and go, and there may also be increases or decreases in the total size of the fauna. All the estimators we explored assume that the spe-cies pool is constant during the sampling period. Even though the sampling period of the Mal-aise trap inventory spanned over three years, we consider this assumption to be justified.

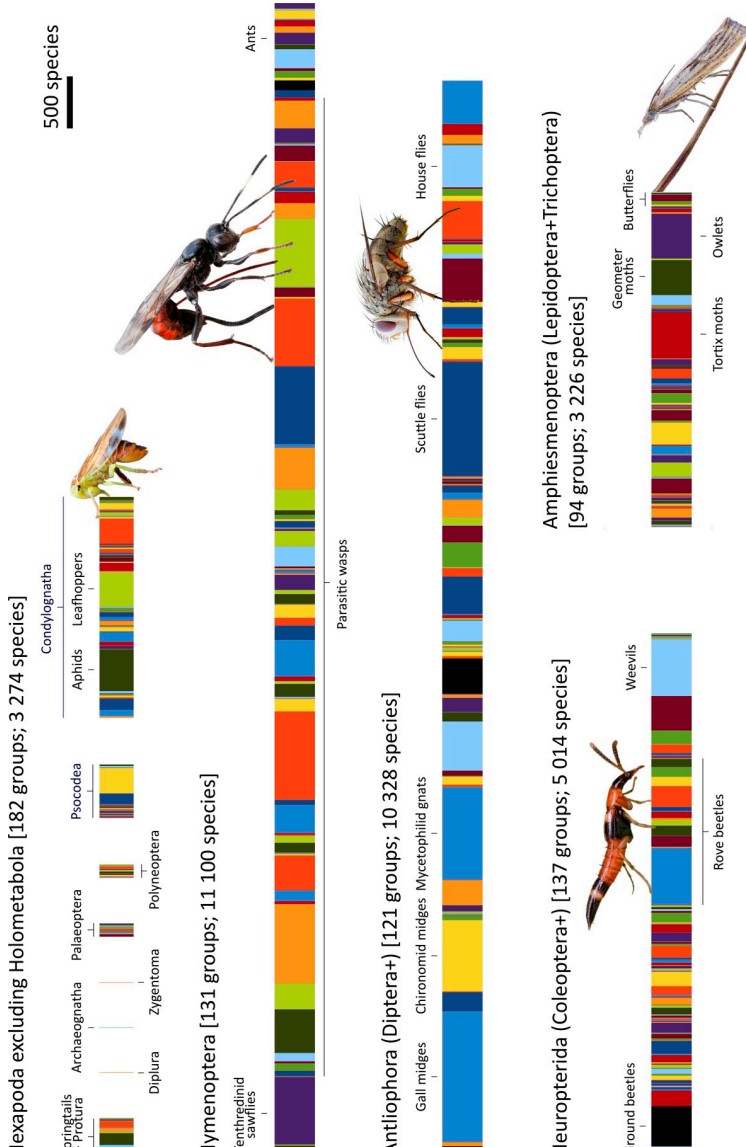

**Fig 6. Detailed view of the taxonomic composition of the true Swedish insect fauna, as estimated here.** Each colored fragment represents a different family (subfamily for Ichneumonidae, Braconidae and Staphylinidae); the width is proportional to the number of species. The families are grouped into monophyletic higher clades according to our current understanding of insect relationships. The inventory has shown that the fauna is much richer in Hymenoptera (sawflies, wasps, ants and bees) and Diptera (midges, gnats, mosquitoes and flies) than expected. These groups vastly outnumber Coleoptera (beetles), Lepidoptera (butterflies and moths) and other insect orders in terms of species diversity. Photographs (CC BY) John Hallmén.

A more serious cause of trouble is that the species richness statistic is as sensitive to rare and hard-to-detect species as it is to common ones [37]. This is one of the main reasons why most methods tend to underestimate actual diversity in practice [30, 33, 34]. Our results clearly show how important it is to account for heterogeneity across sampling sites when estimating the species pool of a large and heterogeneous geographic area. All estimators that are entirely or partly based on variation across sites (Chao2, Jack1, Jack2 and CNE) do considerably better than the estimators that use abundance data from the pooled sample (Chao1, Preston). This suggests that parametric estimators, such as Preston, which can in principle address the

### Taxonomic composition, true fauna

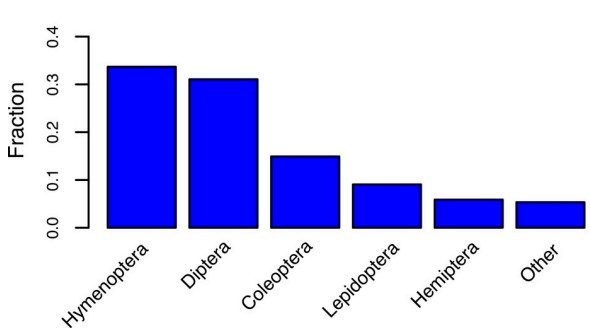

### Taxonomic composition, new species

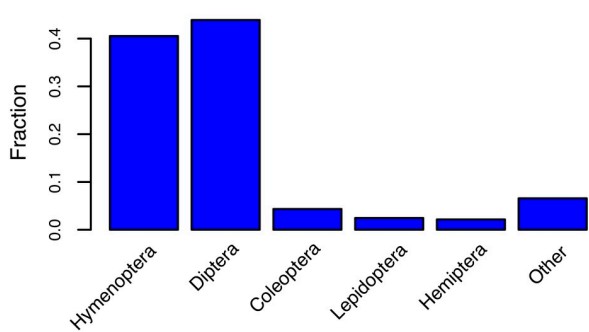

### Niche composition, true fauna

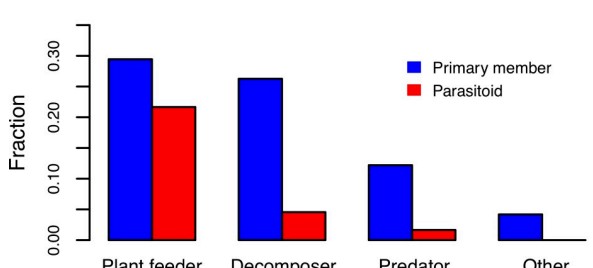

### Niche composition, new species

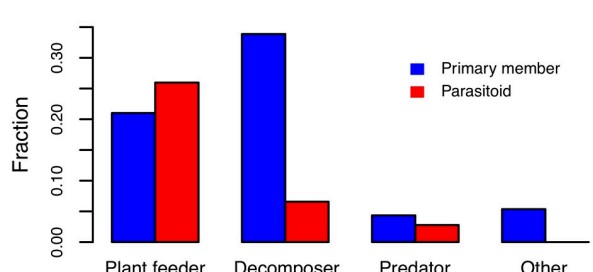

### Habitat composition, true fauna

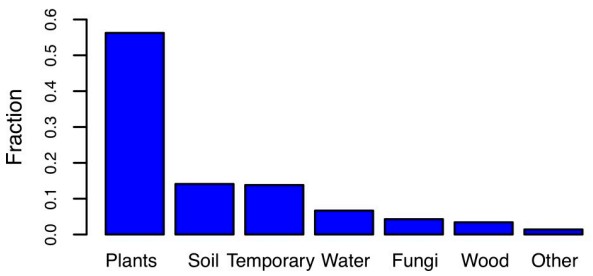

### Habitat composition, new species

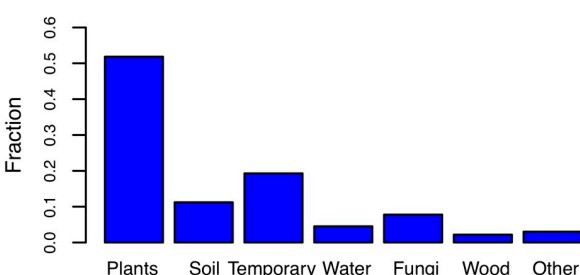

**Fig 7. Composition of the Swedish insect fauna, and changes in our understanding of the composition brought about by the inventory.** The taxonomic and ecological composition of the fauna, as estimated here (33,000 spp.), is given to the left. The composition of the species that were still unknown in 2003 (8,600 spp., many of which still await discovery), as predicted by their phylogenetic affinities (family or subfamily placement), is shown to the right.

unequal abundance of species, need to be developed such that they also address the spatial variation in the frequency of species in the species pool.

Despite the inherent difficulties of estimating species richness, our combined non-parametric estimator (CNE) comes very close to the expert assessments of the diversity in the critical groups for which we have data (Tables 1 and S5). The same is true for the incidence-based estimators that include a correction for low sampling intensity, in particular Chao2P and Jack2P. The advantage of our CNE method over the latter is that it is based on testable assumptions about the causes of underestimation at low sampling intensities.

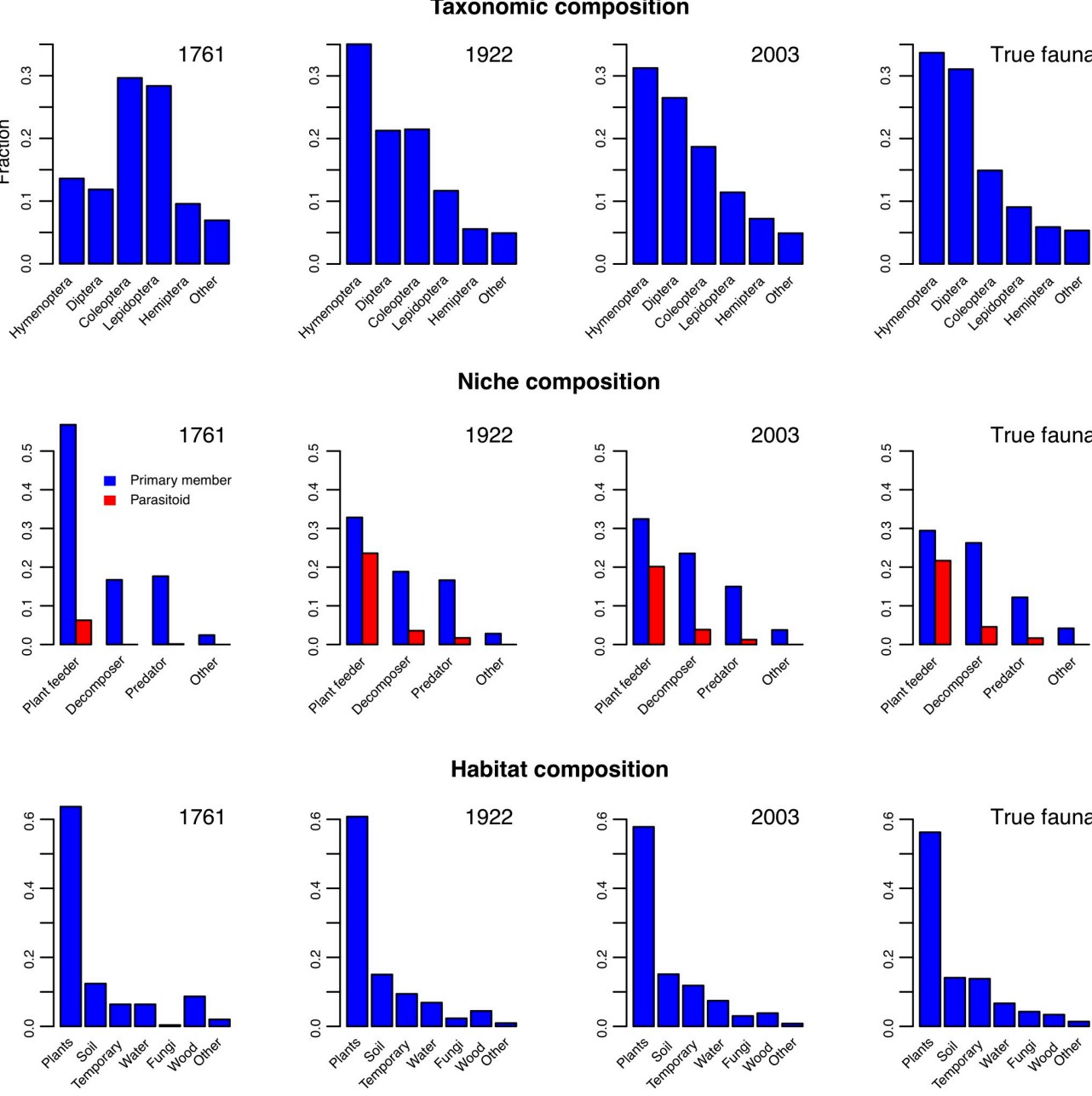

**Fig 8. Changes in apparent composition of the Swedish insect fauna over time as it has become better known.** Over time, the apparent fraction of Hymenoptera and Diptera species has increased dramatically. The same is true for the community of decomposers and parasitoids (red bars). Changes in apparent microhabitat composition have been less pronounced; most insects live on or in plants. However, the proportion of species found in other microhabitats has increased over time.

The mark-recapture estimator (MR) also comes very close to expert opinion about the likely diversity of the critical groups (Tables 1 and S5). It is worth pointing out that this estimate is based on a completely different source of information (the proportion of new species encountered in the sample) than the other estimators considered here. The expert assessments themselves are probably based to some extent also on the proportion of new species encountered in the Malaise trap inventory but they presumably also account for species abundance and incidence data. They are also based on knowledge of the biology and distribution of the

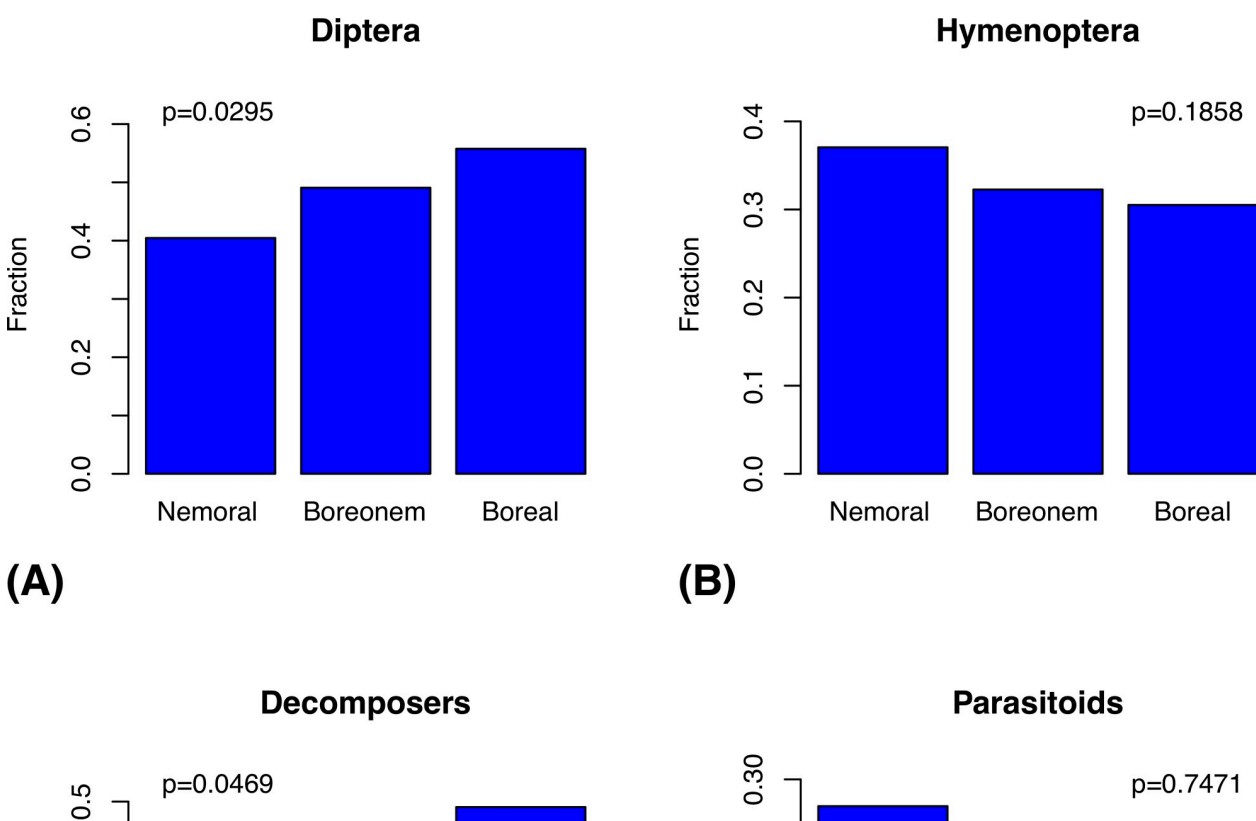

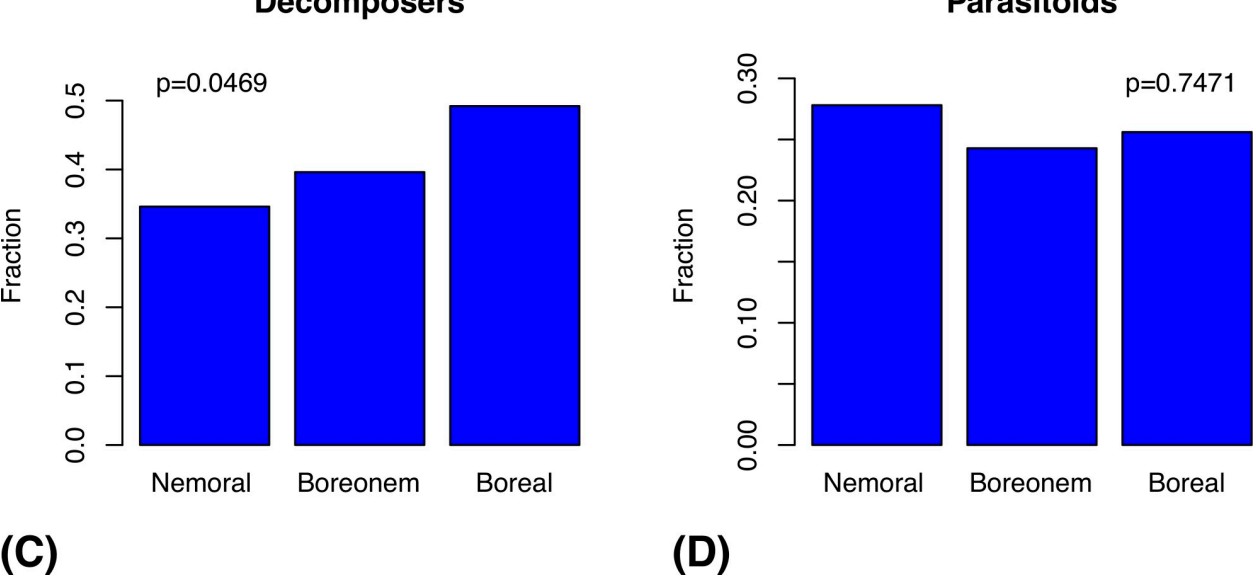

**Fig 9. Compositional changes in the Swedish insect fauna over the nemoral to boreal transition, as indicated by the Malaise trap inventory data.** The plots show the changes in the proportion of species that belong to Diptera (A) and Hymenoptera (B), and the proportion of species that are decomposers (C) and parasitoids (D). The *p* values were computed by comparing the observed values with a reference distribution generated using 10,000 random permutations of the trap assignments to biogeographic region. The arctic zone was excluded from this comparison because there is an order of magnitude less inventory data for this zone, and data are lacking for key groups. Sample sizes: nemoral zone 18 traps, 1,557 species; boreonemoral zone 29 traps, 2,693 species; and boreal zone 22 traps, 1,691 species.

group, special circumstances affecting particular Malaise trap catches, and data from other collecting efforts than the Malaise trap inventory.

In conclusion, several lines of reasoning suggest that our estimate of the Swedish insect fauna represents the most accurate picture yet of the true size and composition of a sizeable temperate insect fauna. First and foremost, the Swedish insect fauna is one of the best known in the world, which reduces the uncertainty in the extrapolations needed to predict the total,

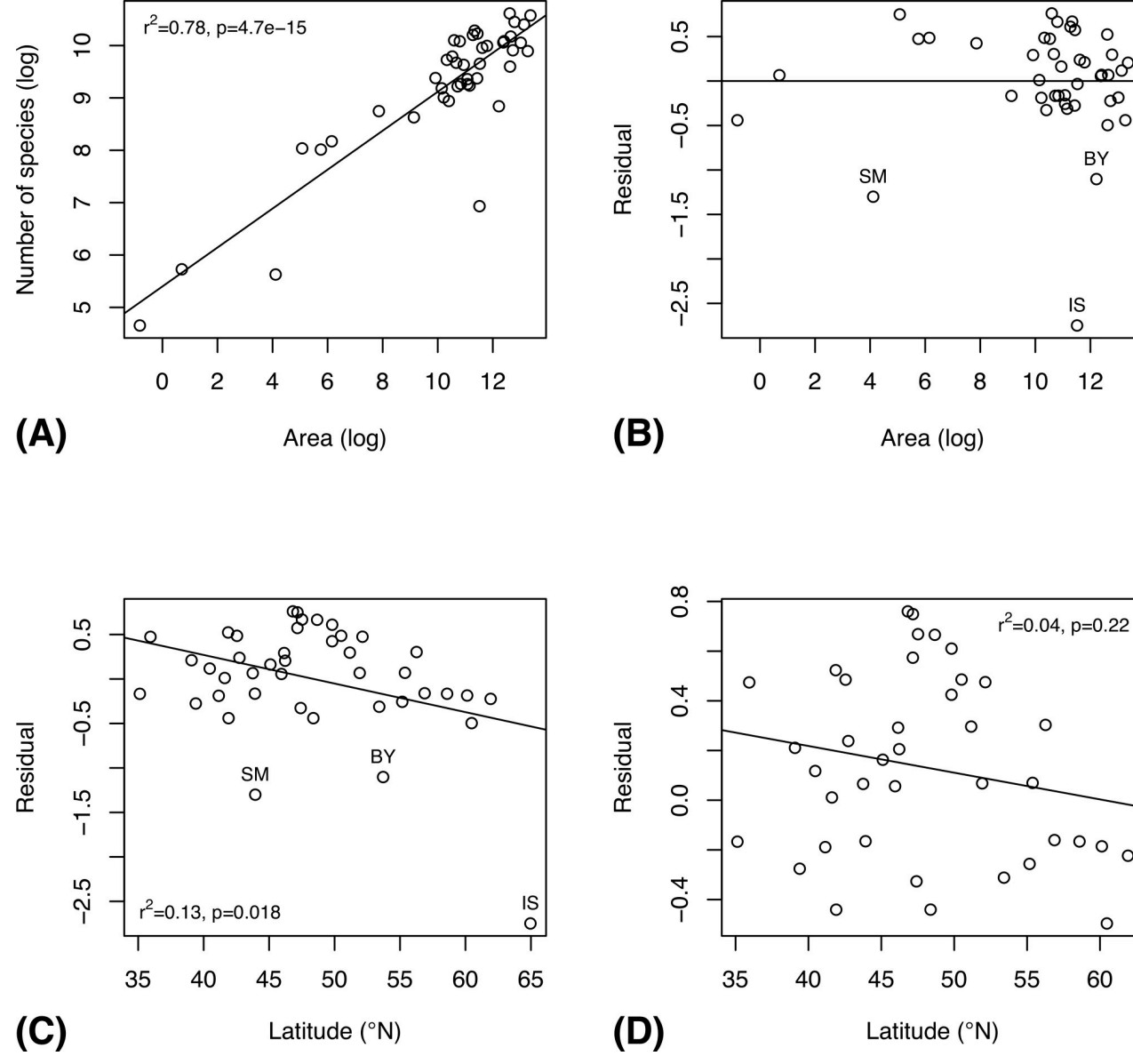

**Fig 10. Latitudinal gradient in overall species richness of national European insect faunas, as documented by the Fauna Europaea database.** (A) Relationship between country area and the species richness of its insect fauna (log-log plot). (B) Residuals from the previous plot. Note three outliers—Belarus (BY), San Marino (SM) and Iceland (IS)—with unexpectedly small insect faunas. (C) Latitudinal gradient in species richness after the area effect is removed, using all data. (D) Ditto, after removal of the three outliers.

and allows us to learn about the strengths and weaknesses of the extrapolation methods using a wide range of groups for which we have reliable data on the total diversity. If we include the species identified in the inventory as new to science but not yet described, approximately 28,000 insect species are now known from Sweden, leaving only 5,000 expected species (15%) that remain to be discovered. This possibly represents the smallest gap between the known and the true species stocks of any major insect fauna studied to date. Second, the Swedish Taxonomy Initiative has funded more than 15 years of focused research on the most poorly known groups in the Swedish insect fauna, precisely the ones that are critical for estimating the size and composition of the real fauna. Finally, the three independent approaches used here to estimate the total generate consistent results, as detailed above.

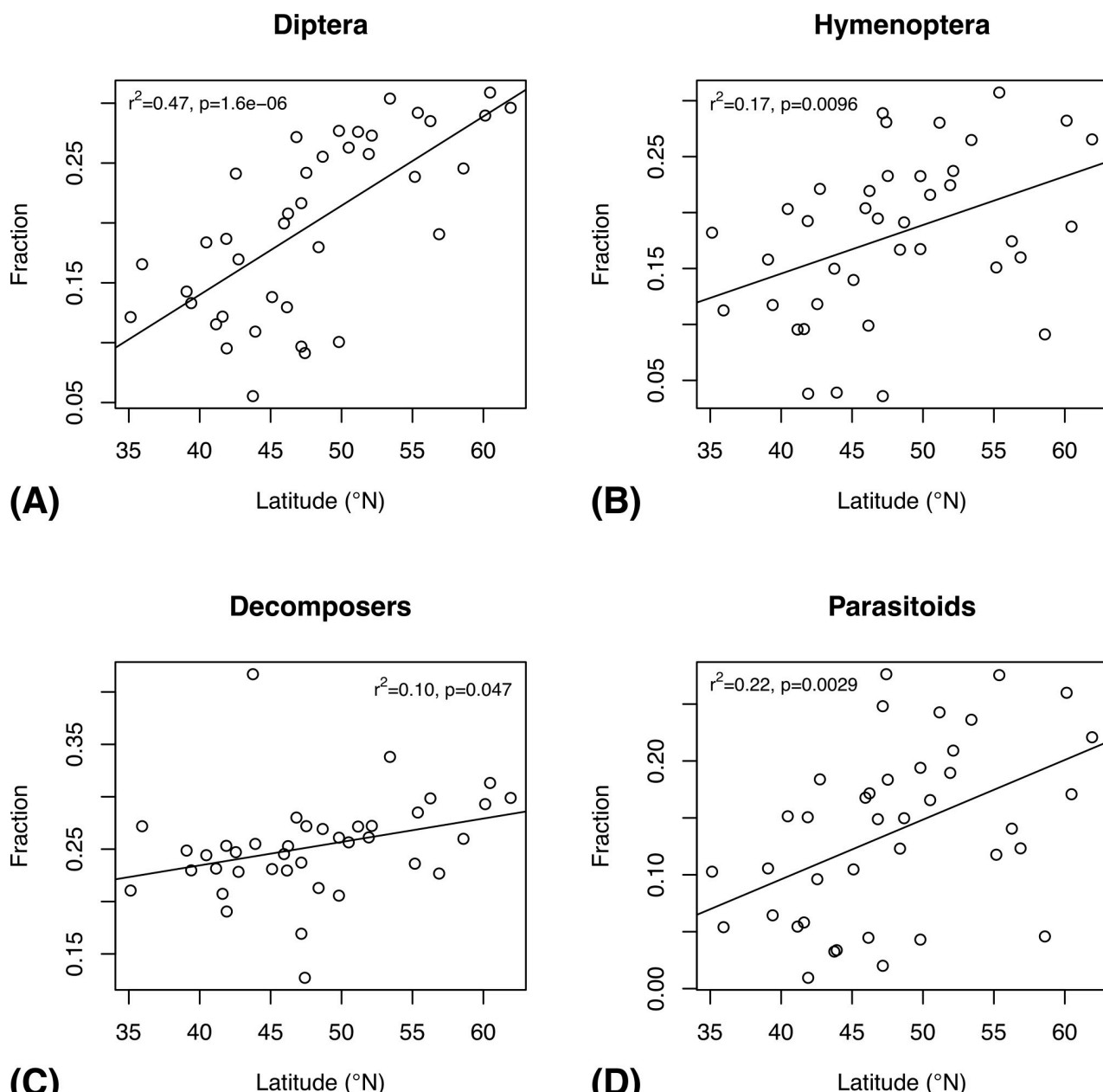

**Fig 11. Latitudinal trends in the composition of national European insect faunas, as documented by the Fauna Europaea database.** The plots show the latitudinal gradients in the proportion of species that belong to Diptera (A) and Hymenoptera (B), and the proportion of species that are decomposers (C) and parasitoids (D). Three outliers (see Fig 10B) were removed from these analyses.

Thus, it appears that we can, for the first time, answer some basal questions about the composition of the fauna with some precision. The answers themselves give rise to new questions. For instance, it appears that the plant feeders and their parasitoids comprise about one half of the fauna, while decomposers and their parasitoids constitute one third. Why is this the case? Why is it that there are fewer parasitoids per decomposer species than there are per plant feeder? Clearly, pursuing the answers to those questions will generate important new insights into the structure and function of the insect ecosystems in Sweden and elsewhere.

Our analysis of historical data clearly shows how dramatically our understanding of the Swedish insect fauna has changed since Linnaeus's times. Even as we appear to approach a complete picture of the fauna, the composition is still changing markedly (Figs 7 (right) and 8). Given this, one may be justified in asking to what extent knowledge biases similar to the ones distorting our view of the Swedish insect fauna in the past may affect our current perception of worldwide patterns in insect faunas. For instance, the Swedish fauna is richer in Hymenoptera and Diptera than in Coleoptera and Lepidoptera, and this has been known to be the case for temperate insect faunas for some time. Tropical insect faunas, however, are famous for their richness in beetles. In the past, this difference has been assumed to be real [38–39], but recent studies suggest that the poor representation of Diptera and Hymenoptera in tropical faunas is more likely due to insufficient study [40–46].

Somewhat surprisingly, there is no significant decrease in species diversity with increasing latitude in known European insect faunas, if Iceland is omitted from the analysis (Fig 10). However, this may well be due to insufficient study of the insect faunas of southern Europe (see also [47]). The Swedish inventory data do show the expected decrease in diversity over the nemoral to boreal transition. The European data reveal an increase in Hymenoptera and parasitoids with increasing latitude (Fig 11), but the Swedish data (Fig 9) suggest that these trends may be artifactual. On the other hand, the Swedish data do support a northern increase in Diptera and decomposers, as documented by the European data.

In recent years, insect faunas have increasingly been studied with DNA barcoding methods. It is interesting to compare our results with those of a recent study of the Canadian insect fauna based on DNA barcoding of insects collected with Malaise traps, pitfall traps, sweep nets, intercept traps and light traps [48]. To our knowledge, this is the most extensive barcoding study of an insect fauna published thus far. The Canadian study agrees with ours in finding extensive undocumented Diptera diversity, more than doubling previous estimates, but fails to detect a similar effect for Hymenoptera. Thus, the Canadian fauna is supposedly much richer in Diptera than in Hymenoptera, in contrast to the inferred Swedish fauna. It is possible that the faunas are indeed different, but we consider it more likely that the Canadian results are misleading because of the low coverage of many critical Hymenoptera taxa and the low success in barcoding hymenopterans (see also [49]). It is true that the Canadian study is based on a larger sample than our study (1,000,000 versus 165,000 specimens). However, the Canadian study also covers a larger geographic area, analyzes a larger and less studied fauna, and lacks the focus on critical taxa characterizing the data reported here. It is also unclear what principles were used to select specimens for barcoding; without specific selection criteria, it does not appear unlikely that many critical taxa of small Hymenoptera and Diptera were underrepresented. Furthermore, most of the genetically circumscribed species in the Canadian study (BINs) lack morphological validation. Thus, we argue that our data are more informative about the real size and composition of the Swedish insect fauna, than the Canadian data are about the Canadian fauna.

It appears unlikely that the apparent difference in taxonomic composition between the Swedish and Canadian insect faunas is due to undetected cryptic diversity of Diptera in Sweden. DNA barcoding tends to be consistent with traditional morphology-based taxonomy when sufficient resources have gone into the latter [50–53]. Barcoding of critical Swedish Diptera taxa (Phoridae, Cecidomyiidae, Chironomidae, and Mycetophilidae) have revealed 0–20% cryptic diversity [36, 54, 55] but similar results have been obtained for parasitic Hymenoptera [56]. If anything, it seems likely that systematic and large-scale DNA barcoding will show that our current estimates for Swedish insect groups are conservative both for Diptera and Hymenoptera.

Metabarcoding, simultaneous DNA barcoding of all specimens in a bulk sample using high-throughput sequencing platforms, promises to revolutionize the inventory of faunas and floras. However, these methods are still in development. Recent results show surprising inconsistencies in metabarcoding results for Malaise trap samples depending on whether the preservative ethanol or tissue homogenate is used [57]. Specifically, it appears that small and weakly sclerotized insects, which are detected in preservative ethanol, are easily missed in tissue homogenate because the signal is swamped by the DNA from larger insects. What is true for tissue homogenate may also be true for the standard lysis protocols used in metabarcoding, which are quite aggressive. For instance, a recent study noted that their metabarcoding protocol failed to pick up a lot of the diversity of small insects that was visually apparent in the sample [58]. They also showed that separate analysis of different taxonomic fractions from the sample revealed more diversity than a single analysis of the pooled sample. Thus, there is more development work needed before the full power of metabarcoding can be effectively brought to bear on questions about the real size and composition of large insect faunas.

Recent years have also seen the development of promising new approaches that reduce the cost of single-specimen DNA barcoding to such an extent that large numbers of specimens can be processed [59–61]. In contrast to metabarcoding, these methods allow individual specimens with deviating genetic signal to be identified and studied by taxonomists post sequencing. This will reduce or even eliminate the need for morphospecies sorting of material intended for taxonomic study, thus accelerating the discovery of cryptic species and enabling rapid processing of thousands of specimens of understudied taxa [62, 63].

What is abundantly clear from this study is that our current understanding of the size and composition of most insect faunas, even those of the temperate region, is seriously biased because of the lack of study of large and important taxonomic and ecological groups (see also [64]). This impedes our understanding of how insect faunas are composed, and our ability to monitor the ecosystem services they provide. Addressing this knowledge gap is all the more urgent as we may possibly be facing a worldwide decline in the abundance and diversity of insects ([65–67]; but see also [68, 69]).

## Supporting information

**S1 Table. Taxonomic and biological data for the Swedish insect fauna.** Data from older sources were reorganized to match the taxonomic classification of DynTaxa in 2017. Yellow cells are commented in the 'Comments' column. Obvious errors detected in Gärdenfors et al. (2003) and in the DynTaxa data from 2017 were corrected, as noted in the comments, and the corrected numbers were used in all analyses.
(XLSX)

**S2 Table. Data on the trap sites and samples from the Malaise trap inventory.** Sheet 1 contains information about the location of the trap sites, the habitat at those sites, and collecting period. Sheet 2 contains information about the collecting events and the corresponding dates, used to compute the total collecting effort represented by the samples that have been processed for different groups.
(XLSX)

**S3 Table. Overview of Malaise trap inventory data files.** The table provides an overview of the raw data files and their content.
(XLSX)

**S4 Table. Overview of the groups used in the species richness estimation.** The analysis groups ('Analysis taxon') match the specimen batches processed by different taxonomic

experts, each corresponding to a different subset of samples and sites for a specific taxon or group of taxa, and therefore do not always match the family-level taxa used in the compositional analyses (S1 Table). The proportion of samples processed of each group was estimated based on data from the sorting of 1481 of the 1919 samples (77% of the entire material). In a few cases, as noted in the table, the sorting fractions did not match the analysis groups, and instead the proportion of the catch was estimated from 100 arbitrarily chosen samples sorted to the appropriate groups.
(XLSX)

**S5 Table. Species richness estimates.** The data are divided into four subsets, presented on different sheets in the file. The first sheet gives the data for the groups where new species have been found in the Malaise trap inventory ('poorly studied groups') and where abundance data are available. The second sheet gives the data for poorly studied groups where only incidence data are available. The third and fourth sheets give the richness estimates based on abundance and incidence data, respectively, for 'well-studied groups', defined as those groups where no new species have been found in the inventory.
(XLSX)

**S6 Table. Taxonomic overview and life-history traits for the Fauna Europaea data.** Life history traits are given for the family-level taxa used in the analysis of the Fauna Europaea data. Note that these family-level taxa occasionally deviate from the family-level taxa used in the analysis of the Swedish data. In the comparative analyses of European insect faunas, the data for the Swedish fauna was taken from Fauna Europaea. These data largely correspond to the knowledge of the Swedish fauna in 2003 as documented in S1 Table; we did not attempt to correct the data.
(XLSX)

**S1 Fig. Accuracy of different estimators in predicting the species richness of well-studied groups, and the species pool known in 2003 of more poorly studied groups.** The plots show the accuracy (measured as squared error of log estimates) of the species richness estimates as a function of the proportion of singletons (A), the proportion of uniques (B), the number of specimens per species (C), and the number of site observations per species (D). A "singleton" is a species represented by a single specimen; a "unique" is a species that only occurs at one site.
(TIFF)

**S2 Fig. Bias in P-corrected species richness estimators as a function of the fraction of the species pool sampled.** We show the bias in four P-corrected species richness estimators: Chao1P (A), Chao2P (B), Jack1P (C) and Jack2P (D). Bias is measured on the log scale; the horizontal line represents absence of bias.
(TIFF)

## Acknowledgments

We would like to thank Kajsa Glemhorn, Pelle Magnusson, and Mareike Kiupel for invaluable assistance in coordinating the Malaise trap inventory. Anders Göthberg, Bengt-Åke Bengtsson, Gerhard Bächli, Gösta Gillerfors (†), Aurel I. Lozan, Emilia Nartshuk, Marko Prous and many additional experts and volunteers contributed in important ways to the inventory. John Hallmén provided the insect photos illustrating the composition of the fauna. Marko Prous, Pedro Cardoso, Pierfilippo Cerretti and an anonymous reviewer provided constructive comments that helped improve the final version of the manuscript.

## Author Contributions

**Conceptualization:** Fredrik Ronquist, Ulf Gärdenfors.

**Data curation:** Mattias Forshage, Sibylle Häggqvist, Dave Karlsson, Rasmus Hovmöller, Kevin Holston, Johan Abenius, Bengt Andersson, Peter Neerup Buhl, Carl-Cedric Coulianos, Arne Fjellberg, Carl-Axel Gertsson, Sven Hellqvist, Mathias Jaschhof, Jostein Kjærandsen, Seraina Klopfstein, Sverre Kobro, Andrew Liston, Rudolf Meier, Marc Pollet, Matthias Riedel, Jindřich Roháček, Meike Schuppenhauer, Julia Stigenberg, Ingemar Struwe, Andreas Taeger, Oleksandr Varga, Phil Withers.

**Formal analysis:** Fredrik Ronquist, Rasmus Hovmöller, Tom Britton.

**Funding acquisition:** Fredrik Ronquist, Dave Karlsson.

**Investigation:** Fredrik Ronquist, Mattias Forshage, Sibylle Häggqvist, Dave Karlsson, Rasmus Hovmöller, Kevin Holston, Johan Abenius, Bengt Andersson, Peter Neerup Buhl, Carl-Cedric Coulianos, Arne Fjellberg, Carl-Axel Gertsson, Sven Hellqvist, Mathias Jaschhof, Jostein Kjærandsen, Seraina Klopfstein, Sverre Kobro, Andrew Liston, Rudolf Meier, Marc Pollet, Matthias Riedel, Jindřich Roháček, Meike Schuppenhauer, Julia Stigenberg, Ingemar Struwe, Andreas Taeger, Sven-Olof Ulefors, Oleksandr Varga, Phil Withers, Ulf Gärdenfors.

**Methodology:** Fredrik Ronquist, Ulf Gärdenfors.

**Project administration:** Fredrik Ronquist, Dave Karlsson.

**Resources:** Fredrik Ronquist.

**Supervision:** Fredrik Ronquist, Dave Karlsson.

**Validation:** Fredrik Ronquist, Mattias Forshage, Sibylle Häggqvist, Dave Karlsson, Kevin Holston.

**Visualization:** Fredrik Ronquist, Rasmus Hovmöller, Johannes Bergsten.

**Writing – original draft:** Fredrik Ronquist, Johannes Bergsten.

**Writing – review & editing:** Fredrik Ronquist, Mattias Forshage, Sibylle Häggqvist, Dave Karlsson, Rasmus Hovmöller, Johannes Bergsten, Kevin Holston, Tom Britton, Johan Abenius, Bengt Andersson, Peter Neerup Buhl, Carl-Cedric Coulianos, Arne Fjellberg, Carl-Axel Gertsson, Sven Hellqvist, Mathias Jaschhof, Jostein Kjærandsen, Seraina Klopfstein, Sverre Kobro, Andrew Liston, Rudolf Meier, Marc Pollet, Matthias Riedel, Jindřich Roháček, Meike Schuppenhauer, Julia Stigenberg, Ingemar Struwe, Andreas Taeger, Sven-Olof Ulefors, Oleksandr Varga, Phil Withers, Ulf Gärdenfors.

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
