## [Decision Letter · Decision Letter 0]

4 Nov 2019

PONE-D-19-26826

Completing Linneaus's inventory of the Swedish insect fauna: only 5000 species left

PLOS ONE

Dear Dr. Ronquist,

Thank you for submitting your manuscript to PLOS ONE. After careful consideration, we feel that it has merit but does not fully meet PLOS ONE’s publication criteria as it currently stands. Therefore, we invite you to submit a revised version of the manuscript that addresses the points raised during the review process.

We would appreciate receiving your revised manuscript by Dec 19 2019 11:59PM. To enhance the reproducibility of your results, we recommend that if applicable you deposit your laboratory protocols in protocols.io, where a protocol can be assigned its own identifier (DOI) such that it can be cited independently in the future. For instructions see: http://journals.plos.org/plosone/s/submission-guidelines#loc-laboratory-protocols

We look forward to receiving your revised manuscript.

Kind regards,

Pierfilippo Cerretti, Ph.D.

Academic Editor

PLOS ONE

Journal Requirements:

Additional Editor Comments (if provided):

Dr. Ronquist,

The review process to manuscript PONE-D-19-26826 have been completed. Two reviewers provided interesting comments and helpful suggestions. The first reviser suggets that the section about the development of a new non-parametric estimator (CNE) should go in a separate paper. Although I agree about the need to highlight and discuss this section more clearly, I think this can be done in this paper. Both reviewers also agree that the manuscript structure is unbablanced, with a too short Discussion section in which only part of the results have been discussed. I think, for instance, that the Results section includes a lot of text which should be moved to the Discussion section.

Should you agree to resubmit a revised version of your manuscript, please provide a point-by-point reply to reviewers' comments.

Best wishes,

Pierfilippo Cerretti

Reviewers' comments:

Reviewer's Responses to Questions

**Comments to the Author**

1. Is the manuscript technically sound, and do the data support the conclusions?

Reviewer #1: Yes

Reviewer #2: Yes

2. Has the statistical analysis been performed appropriately and rigorously? 

Reviewer #1: Yes

Reviewer #2: Yes

3. Have the authors made all data underlying the findings in their manuscript fully available?

Reviewer #1: Yes

Reviewer #2: Yes

4. Is the manuscript presented in an intelligible fashion and written in standard English?

Reviewer #1: Yes

Reviewer #2: Yes

5. Review Comments to the Author

Reviewer #1: My main concern is that the authors tried to say too much in a single paper, including parts, such as the development of new estimators, that could go to their own paper. And many parts are confusing, probably due to trying to do too much in a single go. The extensive amount of figures and tables reflects this. I could see many ways in which the paper could be shortened and streamlined.

As I read the methods before the results (as makes most sense to me, as results without methods are mostly impossible to interpret) the first part of the results sounded a repetition of methods. I advise to swap the order and avoid having two methods sections.

The entire section of the development of the new estimator could be on its own paper. Developing a new estimator based on a limited number of tests might be a step too far. Also, the message might be lost as it is not the main goal of the paper. At some point it gets tiring to read so much information and details on so many different methods. Twice…

Minor comments:

Ln 163 – In the abstract the authors mention 301 species new to science, not 609. Somehow it seems these are different datasets, but it is not clear.

Ln 180 – If these are for all groups for which new species were found, how can they represent 29% of the non-recorded fauna and not 100%? Confusing.

Ln 429 – What is the difference between S. borealis and the next five? I recommend to reduce the length of this part.

Ln 465 – What latitudinal Analysis? And it is Selvagens, not Selagen.

Ln 598 – There is a class of estimators that are often found to be less biased than the ones used, the P-corrected versions of Chao1, Chao2, Jack1 and Jack2. These are available in the R package BAT and could be added.

Ln 692 – I am not sure about what this sentence means. The pool is known because it is defined a priori? Was it defined a priori because it was known? Looks like circular reasoning. Rewrite.

Ln 694 – Which two data sources?

Fig 1 – What are trap-days per taxon? In principle all taxa could fall into all traps. Rename or delete.

Pedro Cardoso

Reviewer #2: Major comments

The result section is very long presenting data and figures about many question and is not balanced compared to the discussion. The discussion is very short focusing on methodological issues and DNA barcoding and the gap of knowledge and the importance of Hymenoptera and Diptera in comparison to tropical regions and the dominance of Coleoptera. I think the discussion should be extended.

Given the information and statement in the manuscript that “15 years of focused research on the most poorly known groups in the Swedish insect fauna! And that only “data from the first 165,000 specimens that have been identified, representing about 1% of the catch” it’s rather depressing that only 1% of the material have been identified during 15 years of work. This naturally comes to the question why no metabarcoding has been done to justify the use of traditional taxonomical identification methods that the authors argue is required. Today metabarcoding is a cheap and well established method to explore species richness in samples that contain thousands or millions of specimens. Lacking such an approach make the study a bit old fashioned and speculative. In the recent literature other studies have come up with contradictory conclusions (among several papers in this topic Morinière et al 2016).

In the material and methods it’s assumed that the fauna is constant over time? How about the current extinction rates that are expected to exceed 1000 times higher than in previous era and the colonisation rates in Sweden that should be 1-5% per year depending on taxonomic group? These topics should be mentioned in the manuscript and if it’s not an issue it should be justified. How many species have been extinct from Sweden already? Is only a fraction of the species occurring there still present? Especially this is important for the “true fauna” in Figure 6. Is there a true balanced fauna in Sweden neutral to extinction/colonisation dynamics? Faunas are in most areas in a continuous change especially in recent years?

Fig. 9-11 is poorly or not at all discussed in the discussion. They deserve a paragraph per figure.

Why are the recent developments in metabarcoding using the NGS platforms ignored in the manuscript? This deserves more space in the manuscript especially since the discussion is focused on DNA. Among many recent studies in the topic:

Reference

Species Identification in Malaise Trap Samples by DNA Barcoding Based on NGS Technologies and a Scoring Matrix

Jérôme Morinière ,Bruno Cancian de Araujo,Athena Wai Lam,Axel Hausmann,Michael Balke,Stefan Schmidt,Lars Hendrich,Dieter Doczkal,Berthold Fartmann,Samuel Arvidsson,Gerhard Haszprunar

ttps://doi.org/10.1371/journal.pone.0155497

Minor comments

Macroscopic diversity – what is this? It is commonly used in other disciplines of science. If the authors refer to “referring to objects that are large enough to be seen easily with the naked eye” I doubt it’s this is the case for many animals this paper cover?

Row 109-111. Country? I don’t understand? It must be easy to calculate the proportion of known species in Sweden of well know groups (birds/plants/Lepidoptera/Bees etc) compared to the species occurring in Europe. It should be about 25% on average if that’s what the sentence is trying to explain. Half is probably an overestimation and require references (if true). Sweden is situated on young terrain with low average temperatures and have an expected very low-low diversity. Only a few cold adapted groups such as Bombus spp are expected to divide from this pattern.

Row141: Collecting?

Row167: Biogeographic breakdown?

Figure one is hard to read, increase resolution/font size

Row 182—185: Reference to the statement “analogous to mark-recapture approximation of

184 population size” Did you use Craigs estimate or Jolly Seber or others? Its hard to find in methods.

Row 275: Habitat breakdown ?

6. PLOS authors have the option to publish the peer review history of their article (what does this mean?). If published, this will include your full peer review and any attached files.

Reviewer #1: Yes: Pedro Cardoso

Reviewer #2: No

---

## [Author Response · Author response to Decision Letter 0]

19 Dec 2019

See separate Response to Reviewers, detailing how we modified the manuscript based on the specific reviewer and editor comments.

---

## [Decision Letter · Decision Letter 1]

21 Jan 2020

Completing Linnaeus's inventory of the Swedish insect fauna: only 5,000 species left?

PONE-D-19-26826R1

Dear Dr. Ronquist,

We are pleased to inform you that your manuscript has been judged scientifically suitable for publication and will be formally accepted for publication once it complies with all outstanding technical requirements.

With kind regards,

Pierfilippo Cerretti, Ph.D.

Academic Editor

PLOS ONE

Additional Editor Comments (optional):

Dear Fredrik,

Both the reviewers and myself think that this is a nice piece of work and I am happy to accept it for publication.

As you probably know, the PLOS ONE team changed the Data Availability statement accompanying this manuscript according to your suggestion.

I am looking forward to seeing it published and hope it will be highlighted in the PLOS ONE webpage.

Congratulations and best wishes,

Pierfilippo Cerretti

Reviewers' comments:

Reviewer's Responses to Questions

**Comments to the Author**

1. If the authors have adequately addressed your comments raised in a previous round of review and you feel that this manuscript is now acceptable for publication, you may indicate that here to bypass the “Comments to the Author” section, enter your conflict of interest statement in the “Confidential to Editor” section, and submit your "Accept" recommendation.

Reviewer #1: All comments have been addressed

Reviewer #2: All comments have been addressed

2. Is the manuscript technically sound, and do the data support the conclusions?

Reviewer #1: Yes

Reviewer #2: Yes

3. Has the statistical analysis been performed appropriately and rigorously? 

Reviewer #1: Yes

Reviewer #2: Yes

4. Have the authors made all data underlying the findings in their manuscript fully available?

Reviewer #1: Yes

Reviewer #2: Yes

5. Is the manuscript presented in an intelligible fashion and written in standard English?

Reviewer #1: Yes

Reviewer #2: Yes

6. Review Comments to the Author

Reviewer #1: All the comments were dealt with and I am happy about the improvements.

Reviewer #2: The revised version has improved substantially. I have no major comments. The word “true” is used frequently >25 times. Sometimes it could be excluded and true could be changed to another word or it could be explained. Is true - the estimated total species richness of insects in Sweden (1700?/2003-2015?)? Are there any true faunas? In Figure 8 for instance it seems there is a constant “true” national fauna in Sweden? True is rarely used in the scientific community the way it is used in the present contribution.

7. PLOS authors have the option to publish the peer review history of their article (what does this mean?). If published, this will include your full peer review and any attached files.

Reviewer #1: Yes: Pedro Cardoso

Reviewer #2: No

---

## [Editor Report · Acceptance letter]

6 Feb 2020

PONE-D-19-26826R1 

Completing Linnaeus's inventory of the Swedish insect fauna: only 5,000 species left? 

Dear Dr. Ronquist:

I am pleased to inform you that your manuscript has been deemed suitable for publication in PLOS ONE. Congratulations! Your manuscript is now with our production department. 

With kind regards,

on behalf of

Dr. Pierfilippo Cerretti 

Academic Editor

PLOS ONE